# ABSTRACT-TO-EXECUTABLE TRAJECTORY TRANSLATION FOR ONE-SHOT TASK GENERALIZATION

## ABSTRACT

Training long-horizon robotic policies in complex physical environments is essential for many applications, such as robotic manipulation. However, learning a policy that can generalize to unseen tasks is challenging. In this work, we propose to achieve one-shot task generalization by decoupling plan generation and plan execution. Specifically, our method solves complex long-horizon tasks in three steps: build a paired abstract environment by simplifying geometry and physics, generate abstract trajectories, and solve the original task by an abstract-to-executable trajectory translator. In the abstract environment, complex dynamics such as physical manipulation are removed, making abstract trajectories easier to generate. However, this introduces a large domain gap between abstract trajectories and the actual executed trajectories as abstract trajectories lack low-level details and are not aligned frame-to-frame with the executed trajectory. In a manner reminiscent of language translation, our approach leverages a seq-to-seq model to overcome the large domain gap between the abstract and executable trajectories, enabling the low-level policy to follow the abstract trajectory. Experimental results on various unseen long-horizon tasks with different robot embodiments demonstrate the practicability of our methods to achieve one-shot task generalization. Videos and more details can be found in the supplementary materials and project page .

## 1 INTRODUCTION

Training long-horizon robotic policies in complex physical environments is important for robot learning. However, directly learning a policy that can generalize to unseen tasks is challenging for Reinforcement Learning (RL) based approaches (Yu et al., 2020; Savva et al., 2019; Shen et al., 2021; Mu et al., 2021). The state/action spaces are usually high-dimensional, requiring many samples to learn policies for various tasks. One promising idea is to decouple plan generation and plan execution. In classical robotics, a high-level planner generates a abstract trajectory using symbolic planning with simpler state/action space than the original problem while a low-level agent executes the plan in an entirely physical environment Kaelbling & Lozano-Pérez (2013); Garrett et al. (2020b). In our work, we promote the philosophy of abstract-to-executable via the learning-based approach. By providing robots with an abstract trajectory, robots can aim for *one-shot task generalization*. Instead of memorizing all the high-dimensional policies for different tasks, the robot can leverage the power of planning in the low-dimensional abstract space and focus on learning low-level executors.

The two-level framework works well for classical robotics tasks like motion control for robot arms, where a motion planner generate a kinematics motion plan at a high level and a PID controller execute the plan step by step. However, such a decomposition and abstraction is not always trivial for more complex tasks. In general domains, it either requires expert knowledge (e.g., PDDL (Garrett et al., 2020b;a)) to design this abstraction manually or enormous samples to distill suitable abstractions automatically (e.g., HRL (Bacon et al., 2017; Vezhnevets et al., 2017)). We refer Abel (2022) for an in-depth investigation into this topic.

On the other side, designing imperfect high-level agents whose state space does not precisely align with the low-level executor could be much easier and more flexible. High-level agents can be planners with abstract models and simplified dynamics in the simulator (by discarding some physical features, e.g., enabling a "magic" gripper Savva et al. (2019); Torabi et al. (2018)) or utilizing an existing "expert" agent such as humans or pre-trained agents on different manipulators. Though imperfect,

their trajectories still contain meaningful information to guide the low-level execution of novel tasks. For example, different robots may share a similar procedure of reaching, grasping, and moving when manipulating a rigid box with different grasping poses. As a trade-off, executing their trajectories by the low-level executors becomes non-trivial. As will be shown by an example soon, there may not be a frame-to-frame correspondence between the abstract and the executable trajectories due to the mismatch. Sometimes the low-level agent needs to discover novel solutions by slightly deviating from the plan in order to follow the rest of the plan. Furthermore, the dynamics mismatch may require low-level agents to pay attention to the entire abstract trajectory and not just a part of it.

To benefit from abstract trajectories without perfect alignment between high and low-level states, we propose **TR**ajectory **TR**anslation (abbreviated as **TR**$^2$), a learning-based framework that can translate abstract trajectories into executable trajectories on unseen tasks at test time. The key feature of **TR**$^2$ is that *we do not require frame-to-frame alignment* between the abstract and the executable trajectories. Instead, we utilize a powerful sequence-to-sequence translation model inspired by machine translation (Sutskever et al., 2014; Bahdanau et al., 2014) to translate the abstract trajectories to executable actions even when there is a significant domain gap. This process is naturally reminiscent of language translation, which is well solved by seq-to-seq models.

We illustrate the idea in a simple Box Pusher task as shown in Fig. 1. The black agent needs to push the green target box to the blue goal position. We design the high-level agent as a point mass which can magically attract the green box to move along with it. For the high-level agent, it is easy to generate an abstract trajectory by either motion planning or heuristic methods. As **TR**$^2$ does not have strict constraints over the

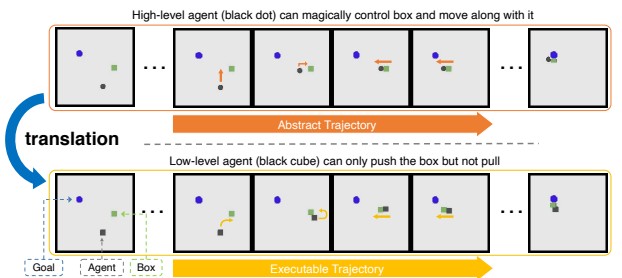

Figure 1: **Task in Box Pusher:** move the green target box to the blue goal position. The arrows in map show how the agents move.

high-level agent, we can train **TR**$^2$ to translate the abstract trajectory, which includes the waypoints to the target, into a physically feasible trajectory. Our **TR**$^2$ framework learns to translate the magical abstract trajectory to a strategy to move around the box and push the box to the correct direction, which closes the domain gap between the high- and low-level agents.

Our contributions are: **(1)** We provide a practical solution to learn policies for long-horizon complex robotic tasks in three steps: build a paired abstract environment (e.g., by using a point mass with magical grasping as the high-level agent), generate abstract trajectories, and solve the original task with abstract-to-executable trajectory translation. **(2)** The seq-to-seq models, specifically the transformer-based auto-regressive model (Vaswani et al., 2017; Chen et al., 2021; Parisotto et al., 2020), free us from the restriction of strict alignment between abstract and executable trajectories, providing additional flexibility in high-level agent design, abstract trajectory generation and helps bridge the domain gap. **(3)** The combination of the abstract trajectory and transformer enables **TR**$^2$ to solve unseen long-horizon tasks. By evaluating our method on a navigation-based task and three manipulation tasks, we find that our agent achieves strong one-shot generalization to new tasks, while being robust to intentional interventions or mistakes via re-planning.

Our method is evaluated on various tasks and environments with different embodiments. In all experiments, the method shows great improvements over baselines. We also perform real-world experiments on the Block Stacking task to verify the capability to handle noise on a real robot system. Please refer to the anonymous project page for more visualizations.

## 2 RELATED WORKS

**One-Shot Imitation Learning** Recent studies (Duan et al., 2017; Finn et al., 2017; Pathak et al., 2018; Yu et al., 2018; Zhou et al., 2019; Yu et al., 2018; Lynch & Sermanet, 2020; Stepputtis et al., 2020) have shown that it is feasible to teach a robot new skills using only a single demonstration, akin to how humans acquire a wide variety of abilities with a single demonstration. To achieve one-shot

generalization, these works usually assume a dataset of expert demonstrations, which is used to train a behavior cloning model (Duan et al., 2017; Rahmatizadeh et al., 2018; Torabi et al., 2018; James et al., 2018) that accelerates learning in future novel tasks. In terms of architecture, Xu et al. (2022) is similar to us but still requires a dataset of low-level demos and is experimented in the offline setting whereas we utilize a dataset of simpler abstract trajectories and train online in complex environments.

**Cross-Morphology Imitation Learning** When there is a morphology difference between expert and imitator, a manually-designed retargeting mapping is usually used to convert the state and action for both locomotion (Peng et al., 2020; Agrawal & van de Panne, 2016) and manipulation (Suleiman et al., 2008; Qin et al., 2022; Antotsiou et al., 2018). However, the mapping function is task-specific, which limits the application of these approaches to a small set of tasks. To overcome this limitation, action-free imitation is explored by learning a dynamics model (Torabi et al., 2018; Radosavovic et al., 2020; Liu et al., 2019; Edwards et al., 2019) to infer the missing action or a reward function (Aytar et al., 2018; Zakka et al., 2022; Sermanet et al., 2016) to convert IL to a standard RL paradigm. However, these methods need extensive interaction data to learn the dynamics model or update policy with learned rewards. Instead of learning the reward function from cross-morphology teacher trajectories, we propose a generic trajectory following reward based on the given abstract trajectory that can provide dense supervision for the agent. DeepMimic (Peng et al., 2018) is similar to ours, but it differs in that we do not allow training at inference time with new trajectories . SILO (Lee et al., 2019) is another similar approach that trains an agent to follow the demonstrator but is limited by fixed window sizes/horizon parameters that inhibit the test generalizability of its approach.

**Demo Augmented Reinforcement Learning** Motion primitives, especially dynamic motion primitives, have long been used by robotics researchers to combine the human demonstration with RL (Kober & Peters, 2009; Theodorou et al., 2010; Li et al., 2017; Singh et al., 2020). Recently, (Pertsch et al., 2021; 2020) extended this framework to learn task-agnostic skills from demonstrations. Another line of work (Ho & Ermon, 2016; Rajeswaran et al., 2017) directly use demonstrations as interaction data. For example, Demo Augmented Policy Gradient (Rajeswaran et al., 2017; Radosavovic et al., 2020) performs behavior cloning and policy gradient interchangeably with a decayed weight for imitation in on-policy training while other works (Vecerik et al., 2017; Hester et al., 2018) append the demonstrations into the replay buffer for off-policy RL. However, their works typically utilize low-level demonstrations while we utilize abstract trajectories that are much easier to generate.

## 3 METHOD

### 3.1 OVERVIEW AND PRELIMINARIES

Similar to one-shot imitation learning (Duan et al., 2017), we are tackling the problem of one-shot task generalization. In one-shot imitation learning, an agent must solve an unseen task given a demonstration (e.g., human demo, low-level demo) without additional training. However, even a single demonstration can be challenging to produce, especially for complex long-horizon robotics tasks. Different from one-shot imitation learning, we replace the demonstration with an abstract trajectory. The abstract trajectory is a sequence of high-level states corresponding to a high-level agent that instructs the low-level agent on how to complete the task at a high level. In the high-level space, we strip low-level dynamics and equip the high-level agent with magical grasping, allowing it to manipulate objects easily. The simplification makes abstract trajectories easier and more feasible to generate for long-horizon unseen tasks compared to human demos or low-level demos.

Given a novel task, our method seeks to solve it with the following three steps: (i) construct a paired abstract environment that can be solved with simple heuristics or planning algorithms and generate abstract trajectories (Sec. 3.2); (ii) translate the high-level abstract trajectory to a low-level executable trajectory with a trained trajectory translator in a closed loop manner (Sec. 3.3, 3.4); (iii) solve the given task and potentially other unseen tasks with the trajectory translator (Sec. 3.5). The three steps above enable our approach to tackle unseen long-horizon tasks that are out-of-distribution as well. Moreover, we can utilize the re-planning feature (regenerating the abstract trajectory during an episode) to increase the success rate at test time to handle unforeseen mistakes or interventions.

Next, we introduce definitions and symbols. We consider an environment as a Markov Decision Process (MDP) $(\mathcal{S}^L, \mathcal{A}, \mathcal{P}, \mathcal{R})$, where $\mathcal{A}$ is the action space, $\mathcal{P}$ is the transition function of the environment, and $\mathcal{R}$ is the reward function. Different from the regular MDP, we consider two state

spaces, the low-level state space $\mathcal{S}^L$ and a high-level state space $\mathcal{S}^H$. We assume there exists a map $f : \mathcal{S}^L \rightarrow \mathcal{S}^H$ and a dissimilarity function $d : \mathcal{S}^L \times \mathcal{S}^H \rightarrow \mathbb{R}$ such that $d(s^L, s^H) = 0$ for $s^H = f(s^L)$, where $s^H \in \mathcal{S}^H$ and $s^L \in \mathcal{S}^L$. The high-level agent generates an abstract trajectory $\tau^H = (s_1^H, s_2^H ..., s_T^H)$ from an initial state $s_1^H = f(s_1^L)$. Note that actions are not included in $\tau^H$. Lastly, the low-level agent receives observations $s_t^L$ in the low-level state space $\mathcal{S}^L$, and takes actions $a_t$ in the action space $\mathcal{A}$, and the dynamics $\mathcal{P}$ returns the next observation $s_{t+1}^L = P(a_t|s_t^L)$.

## 3.2 ABSTRACTING ENVIRONMENTS AND GENERATING ABSTRACT TRAJECTORIES

The first step to solving a challenging task is to build a paired task that abstracts the low-level physical details away. The paired task should be much simpler than the original task so that it can be solved easily. We leverage two general ways to build the abstract environment: (i) simplify geometry (Manolis Savva* et al., 2019), e.g., representing the agent and objects with point masses; (ii) abstract contact dynamics (Srivastava et al., 2021; Kolve et al., 2017), e.g., the original environment requires detailed physical manipulation to grasp objects while the agent in the abstract environment can magically grasp the objects. Thus, leveraging the two methods above, a concrete solution to construct abstract environments for many difficult tasks can be done.

Concretely for all our abstract environments we remove all contact dynamics and enable magical grasp, as well as represent all relevant objects as point masses. The point mass representation further makes the mapping function $f$ simple to define. As a result, making the abstract environment is scalable as we use the same simple process for all environments. This then enables simple generation of abstract trajectories with heuristics with a point mass as the high-level agent. To generate abstract trajectories, for manipulation tasks, we make the high-level agent approach the object, then magically grasp the object, and then finally move the object to the target position. If there are no objects to be manipulated in the task, then it is a simple navigation sequence.

## 3.3 ABSTRACT-TO-EXECUTABLE TRAJECTORY TRANSLATOR

To translate abstract trajectories into executable low-level actions, we aim to learn a low-level policy $\pi_\theta^L(a_t^L|s_t^L, s_{t-1}^L..., s_{t-k+1}^L, \tau^H)$ that utilizes the current and past $k$ low-level states and an abstract trajectory $\tau^H$ generated by a high-level agent. An overview of our translation model and the flow of inputs and outputs is shown in Fig. 2.

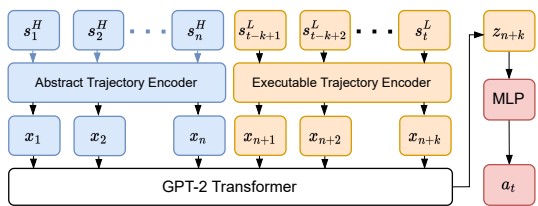

Figure 2: Illustration of the abstract-to-executable trajectory translation architecture. High-level states $s_{H,i}$ are fed through one encoder and the low-level states $s_{L,i}$ are fed through a separate encoder to create tokens. The tokens form a sequence that is given to the transformer model, and the final output embedding $z_{n+k-1}$ is passed through an MLP to produce actions.

The environment at time step $t$ returns to the low-level agent the current and past low-level states $s_t^L, s_{t-1}^L..., s_{t-k+1}^L$ for context size $k$. The high-level agent further provides an abstract trajectory $\tau^H = (s_1^H, s_2^H, ..., s_n^H)$ composed of high-level states $s_i^H$ which can also be viewed as a prompt. However, due to the high-level nature of the abstract trajectory and lack of a frame-to-frame correspondence, learning to follow the abstract trajectory requires a deeper understanding about the abstract trajectory. For more details on the architecture see section E

Thus, we adopt the transformer architecture, specifically GPT-2 (Radford et al., 2019; Brown et al., 2020; Radford et al., 2018), and we format the input sequence by directly appending the current low-level state and past low-level states to the abstract trajectory. With the transformer's attention mechanism (Vaswani et al., 2017), when processing the current low-level state $s_t^L$, the model can attend to past low-level states as well as the entire abstract trajectory to make a decision. By allowing attention over the entire abstract trajectory, our model is capable of modelling long-horizon dependencies better and suffers less from information bottlenecks. For example, in Box Pusher (Fig. 1) the low-level agent must look far ahead into the future in the abstract trajectory to determine which direction the high-level agent (black dot) moves the green target box. By understanding where the target box moves, the low-level agent can execute the appropriate actions to position itself in a way to move the target box the same way and follow the abstract trajectory.

Note that, while the backbone of the model discussed here is the GPT-2 transformer, it can easily be replaced by any other seq-to-seq model like an LSTM (Hochreiter & Schmidhuber, 1997).

### 3.4 Training with Trajectory Following Reward

Conventionally, seq-to-seq models are trained on large parallel corpora for translation tasks like English to German (Luong et al., 2015) in an auto-regressive / open loop manner. However, we desire to train a policy network that solves robotic environments well. To this end, we adapt the seq-to-seq model to a closed-loop setting where the model receives environment feedback at every step as opposed to an open-loop setting, reducing error accumulation that often plagues pure offline methods. Thus, to train the translation model described in Sec. 3.3, we use online RL to maximize a trajectory following reward (1), and specifically, we use the PPO algorithm (Schulman et al., 2017). Note that our framework is not limited to any particular algorithm, and our framework can also work in offline settings if an expert low-level dataset is available.

The core idea of the trajectory following reward is to *encourage the low-level agent to match as many high-level states in the abstract trajectory as possible*. We say a low-level state $s^L$ matches a high-level state $s^H$ when $s^H$ has the shortest distance to $s^L$, and this distance is lower than a threshold $\epsilon$. During an episode, we track the farthest high-level state the low-level agent has matched and use $j_t$ to denote the index of the farthest high-level state which has been matched at timestep $t$.

Concretely $j_t = \max_{1 \le k \le t} j'_k$, where $j'_k = \arg\min_{1 \le i \le n} \{ d(s^L_k, s^H_i) | d(s^L_k, s^H_i) < \epsilon \}$ is the index of the high-level state which is matched by the low-level state $s^L_k$, $n$ is the length of the abstract trajectory, and $d(s^L_t, s^H_{j'}) = ||f(s^L_t) - s^H_{j'}||$. We define our trajectory following reward as follows:

$$
R^{Traj} = \begin{cases} 0 & \text{if } j_t < n \text{ and } j'_t \le j_{t-1} \text{ (make no progress)} \\ (1 + \beta \cdot j'_t) \cdot r_{dist}(s^L_t, s^H_{j'_t}) & \text{if } j_t < n \text{ and } j'_t > j_{t-1} \text{ (make progress)} \\ r_{dist}(s^L_t, s^H_n) & \text{if } j_t = n \text{ (has matched all high-level states)} \end{cases} \tag{1}
$$

Here, $r_{dist}(s^L_t, s^H_{j'}) = 1 - \tanh(w \cdot d(s^L_t, s^H_{j'}))$ is a common distance-based reward function which maps a distance to a bounded reward, and the weight term $(1 + \beta \cdot j'_t)$ is used to emphasise more on the later, more difficult to reach, high-level states. $w$ and $\beta$ are scaling hyperparameters but are kept the same for all environments and experiments. For a visual reward trace of the trajectory following reward function see Sec. C.3 of the appendix.

In practice, we combine the trajectory following reward with the original task reward. Note that the original task rewards are simplistic and not always advanced enough such that a goal-conditioned policy can solve the task. Details about the task reward are in the appendix.

Furthermore, a limitation of the reward function is that it cannot handle abstract trajectories when a high-level state is repeated. This presents a problem in periodic tasks such as pick and placing a block between two locations repeatedly. A simple solution is to chunk abstract trajectories such that each subsequence does not have repeated high-level states and we show an example of this working in section C.2

### 3.5 Test with Trajectory Translation and Re-Planning

At test time starting with low-level state $s^L_1$, we generate an abstract trajectory $\tau^H$ from the mapped initial high-level state $s^H = f(s^L_1)$. At timestep $t$, we execute our low-level agent $\pi^L_\theta$ in a closed-loop manner by taking action $a_t = \arg\max_{a \in \mathcal{A}} \pi^L_\theta(a | s^L_t, s^L_{t-1}..., s^L_{t-k+1}, \tau^H)$. In addition, we are able to re-generate the abstract trajectory in the middle of completing the task to further boost the test performance and handle unforeseen situations such as external interventions or mistakes. We refer to this strategy as *re-planning* and investigate it in Sec. 4.6. Note that re-planning is not adopted in most one-shot imitation learning methods because low-level or human demonstrations are challenging and time-consuming to generate for new tasks and are impractical for long-horizon tasks. In contrast, with a high-level agent acting in a simple high-level state space, we can alleviate these problems and quickly generate abstract trajectories to follow at test time.

In practice, we run re-planning in one of the two scenarios: 1) If the agent matches the final high-level state in $\tau^H$, we will re-plan to begin solving the next part of a potentially long horizon task as done

for Block Stacking and Open Drawer test settings. 2) If after maximum timesteps allowed, the agent has yet to match the final high-level state, then we will re-plan as the agent likely had some errors. Re-planning enables the low-level policy to solve tasks even when there is some intervention or mistake, in addition to allowing it to solve longer-horizon tasks.

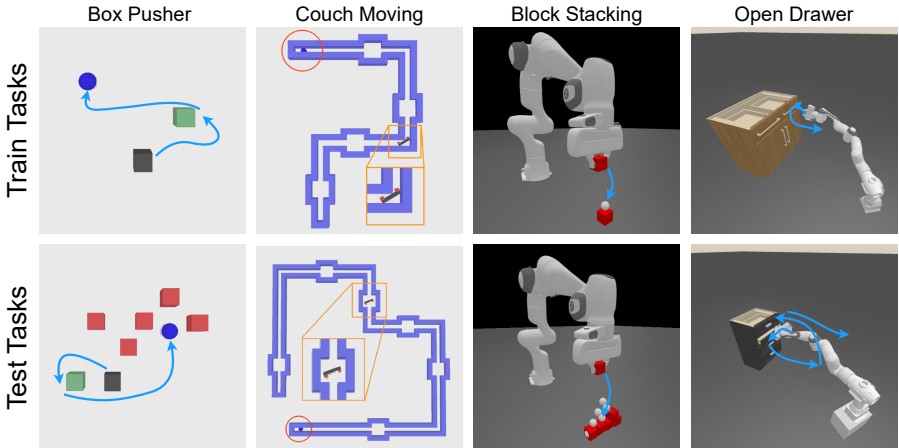

Figure 3: Training and Test Environments, with arrows indicating the direction of movement

## 4 EXPERIMENTS

The effectiveness of $\mathbf{TR}^2$-GPT2 (our $\mathbf{TR}^2$ method with a GPT2 backbone) originates from two key designs: abstract trajectory setup and the complimenting transformer architecture. These two designs contribute to strong performances, especially for long-horizon and unseen tasks.

To evaluate our approach, our experiments will answer the following questions: **(1)** How does $\mathbf{TR}^2$-GPT2 performs compared to other baselines? (Sec. 4.2) **(2)** How does $\mathbf{TR}^2$-GPT2 perform on long-horizon unseen tasks? (Sec. 4.3) **(3)** How does the abstract trajectory setup impact learning? (Sec. 4.4) **(4)** How does $\mathbf{TR}^2$-GPT2 translate trajectories to bridge the domain gap via attention? (Sec. 4.5) **(5)** How can re-planning improve performance in long-horizon tasks? (Sec. 4.6)

To answer these questions, we build four robotic tasks in the SAPIEN (Xiang et al., 2020) simulator with realistic low-level dynamics shown in Fig. 3. These environment support flexible configurations that can generate task variants with different horizon lengths. The Box Pusher task is the simplest and can be used for fast concept verification. The Couch Moving task can test long-horizon dependency and abstract trajectory length generalization. The last two tasks, Block Stacking and Open Drawer, have full-physics robot embodiment with visual sensory input, which can evaluate the performance and generalizability of our method on more difficult high-dimensional robotics tasks. For each environment, we build a paired environment with pointmass-based simplifications and magical grasping to generate abstract trajectories via heuristic methods. See appendix B for more details.

We compare our method, namely $\mathbf{TR}^2$-GPT2, with three baselines on our four environments. $\mathbf{TR}^2$-LSTM: Similar to our method, but replaces GPT-2 with an LSTM (Hochreiter & Schmidhuber, 1997); **Goal-conditioned Policy (GC)**: Instead of using an abstract trajectory as guidance, it receives a task-specific goal and the original task reward, implemented as an MLP; **Subgoal-conditioned Policy (SGC)**: Similar to GC, but it receives a sub-goal, which is a high-level state $n$ steps ahead of the current furthest matched high-level state, and is trained with the same reward as $\mathbf{TR}^2$-GPT2. The correspondence between low- and high-level state are matched using the algorithm in Sec. 3.4.

Furthermore, we compare against a similar line of work, SILO (Lee et al., 2019), and evaluate our method on two of their environments: Obstacle Push and Pick and Place (from SILO).

### 4.1 ENVIRONMENT DETAILS

In this section we outline all experimented environments, detailing the train and test tasks. See Appendix B for additional visuals for all environments including those from SILO.

| | Task | $\text{TR}^2$-GPT2 | $\text{TR}^2$-LSTM | SGC | GC |
|---|---|---|---|---|---|
| 1 | Box Pusher (Train) | 78.7±3.6 | 48.2±4.4 | 32.8±4.2 | **81.3±3.5** |
| 2 | Box Pusher w/ Obstacles | **65.6±4.2** | 29.4±4.0 | 15.1±3.2 | 55.2±4.4 |
| 3 | Couch Moving Short 3 (Train) | **89.6±1.6** | 52.9±4.4 | 41.7±4.4 | 8.6±1.4 |
| 4 | Couch Moving Long 3 | **88.5±1.6** | 52.9±4.4 | 22.9±3.7 | 0.8±0.4 |
| 5 | Couch Moving Long 4 | **74.2±2.2** | 32.8±4.2 | 17.5±3.4 | 0.8±0.4 |
| 6 | Couch Moving Long 5 | **60.7±2.5** | 23.7±3.8 | 11.2±2.8 | 0±0 |
| 7 | Pick and Place (Train) | **95.3±1.9** | 0±0 | 39.0±4.3 | 0±0 |
| 8 | Stack 4 Blocks | **99.0±1.8** | 0±0 | 0±0 | 0±0 |
| 9 | Stack 5 Blocks | **63.5±8.5** | 0±0 | 0±0 | 0±0 |
| 10 | Stack 6 Blocks | **28.1±8.0** | 0±0 | 0±0 | 0±0 |
| 11 | Build 3-2-1 Pyramid | **90.6±5.2** | 0±0 | 0±0 | 0±0 |
| 12 | Build 4-3-2-1 Pyramid | **70.8±8.0** | 0±0 | 0±0 | 0±0 |
| 13 | Open 1 Drawer (Train) | **92.2±2.3** | 52.1±4.5 | 68.5±4.1 | 0±0 |
| 14 | Open 1 Unseen Drawer | **91.4±2.5** | 47.9±4.5 | 62.9±4.3 | 0±0 |
| 15 | Open 2 Drawers | **17.7±6.5** | 15.7±6.4 | 10.4±5.4 | 0±0 |

Table 1: Mean success rate and standard error of training and test tasks, evaluated over 3 training seeds and 128 evaluation episodes each. $\text{TR}^2$-GPT2 outperforms other baselines, especially on test scenarios. Due to a fairly difficult success metric, both $\text{TR}^2$-LSTM and GC could only partially solve block stacking but never completely and GC could only partly solve Open Drawer.

**Box Pusher:** The goal is to control a black box to push a green box towards a blue goal. The low-level agent is restricted to only pushing the green box whereas the high-level agent can magically grasp the green box and pull it along the way. In training there are no red obstacles but these are added at test time. The low-level agent will fail if it cannot follow the abstract trajectory precisely.

**Couch Moving:** The goal is to control a couch shaped agent to complete a maze composed of chambers and corners. The couch shape forces the low-level agent to rotate in chambers ahead of time in order to turn around corners, akin to the moving-couch problem. On the other hand the high-level agent is a pointmass that can easily traverse the entire maze. The naming convention for these tasks is Couch Moving [Short/Long] $n$ where Couch Moving Short 3 is the training task. Short/Long represents the lengths between chambers and corners (Long is about 1.5x longer than Short), and $n$ is the number of corners the agent must turn. The low-level agent can only observe itself and a local patch around it, but does not have direct access to the full maze configuration. As a result, the test settings test whether the low-level agent can process abstract trajectories to determine when to rotate as well as generalize to longer abstract trajectory lengths for test mazes.

**Block Stacking:** The goal is to control a 8-DoF Panda robot arm (w/ gripper) to pick up a block, place it at a goal position, and then return to a resting position. During training, the agent only needs to pick and place a single block up to a height of 3 blocks. At test time, the agent needs to deal with multiple blocks. This setting tests whether the low-level agent can follow out-of-distribution abstract trajectories not seen during training, e.g. stacking higher, placing further away. It also evaluates the performance on much longer horizon tasks.

**Open Drawer:** The goal is to control a 13-DoF mobile robot (w/ gripper) to open a drawer on various cabinets. During training the task is to pull open a drawer on cabinets from a training set. At test time, the task is to pull open drawers on unseen cabinets and / or open additional drawers in an episode. The test setting tests whether the low-level agent can learn to follow the abstract trajectory and manipulate and pull unseen drawer handles.

**Obstacle Push and Pick and Place from SILO (Lee et al., 2019):** We recreate the two tasks from SILO (they did not release code) and compare our method against SILO on these tasks.

## 4.2 RESULTS AND ANALYSIS

As shown in Table 1, our $\text{TR}^2$-GPT2 performs better than all other baselines, especially on test tasks that are unseen and long-horizon. The performance can be attributed to the abstract trajectory setup of $\text{TR}^2$ and the transformer architecture, which will be further investigated in ablation studies.

The GC baseline cannot solve most tasks as the designed task rewards do not provide sufficient guidance. For the SGC baseline, even when conditioned with sub-goals it still has low success rates on test tasks, indicating that simple heuristics are not sufficient to select good subgoals.

| Task | $\mathbf{TR}^2$-GPT2 | SILO |
|---|---|---|
| Obstacle Push | 95.3±1.1 | 70.0 |
| Pick and Place | 100±0 | 95.0 |

Table 2: Results compared with Lee et al. (2019)

The $\mathbf{TR}^2$-LSTM baseline also performs worse than $\mathbf{TR}^2$-GPT2. One interpretation is that modelling long-horizon dependencies with LSTMs is challenging due to the information bottleneck. This prompts us to investigate how the transformer's attention module intuitively leverages the long-term information in Sec. 4.5.

Lastly, compared to SILO, which seeks to imitate a demo as closely as possible like us, we achieve better results shown in Table 2.

## 4.3 Performance on Long-Horizon Unseen Tasks

In Couch Moving, we test the ability of the model to generalize to long-horizon unseen tasks with out-of-distribution abstract trajectories. Rows 3-6 in Table 1 demonstrate how the model can successfully solve much longer and varying mazes, showcasing the expressive power of $\mathbf{TR}^2$-GPT2 compared to baselines. These experiments also show that our model can handle variable sequence lengths and horizons at test time. In comparison, due to fixed/manually tuned horizon sizes, SILO and HRL methods such as HAC have difficulty handling the variance in abstract trajectory lengths after our preliminary attempts.

In Block Stacking, we test the generalizability of the $\mathbf{TR}^2$-GPT2 to handle out-of-distribution task settings, e.g., stacking to higher heights or further locations. Rows 7-12 in Table 1 showcases how our method can stack blocks up to *two times as tall as the training setting*. It can even build new configurations such as a 4-3-2-1 pyramid. As shown in Fig. 6, $\mathbf{TR}^2$-GPT2 can go as far as stacking 26 blocks to build a castle configuration in real-world experiments.

Lastly, in Open Drawer we test the ability of $\mathbf{TR}^2$-GPT2 to generalize to opening drawers of different geometries. Row 14 of Table 1 shows that $\mathbf{TR}^2$-GPT2 can generalize across different drawer handles and reach handles that are farther away compared to the training setting.

## 4.4 How does the granularity of the abstract trajectory impact learning

We perform an ablation study where we vary the granularity of the abstract trajectory, with the most sparse settings resembling closer to that of single-goal conditioned policies and denser settings resembling that of imitation learning. We decrease the granularity by skip sampling more of the abstract trajectory, a process detailed in E, resulting in shorter/sparser abstract trajectories. Results in Fig. 5 (right) show as the abstract trajectory becomes less granular (more sparse), the more difficult it is for agents to learn to follow the abstract trajectory and solve the task. In the Couch Moving task for example, insufficient number of high-level states makes the problem ambiguous and more difficult to determine the correct orientation to rotate into. Thus, low-level policies trained with sufficiently granular high-level states can be successful.

## 4.5 Attention Analysis of $\mathbf{TR}^2$-GPT2 for Bridging the Domain Gap

In general, $\mathbf{TR}^2$-GPT2 will learn whatever is necessary to bridge the domain gap between the high-level and low-level spaces and fill in gaps of information that are excluded from the high-level space. For example, in Couch Moving, the abstract trajectory does not include information about when and how to rotate the couch, only a coarse path to the goal location. Thus the low-level policy must learn to rotate the couch appropriately to go through the corners and follow the abstract trajectory. To visually understand how $\mathbf{TR}^2$-GPT2 learns to bridge the domain gap, we investigate the attention of the transformer when solving the Couch Moving task.

We observe that after training, $\mathbf{TR}^2$-GPT2 exhibits an understanding of an optimal strategy to determine when to rotate or not. As shown in Fig. 4, whenever the agent is in a chamber which permits rotation, the agent attends to positions between the next chamber or next next chamber, all of which are indicative of the orientation of the upcoming corner. Attending to these locations enables

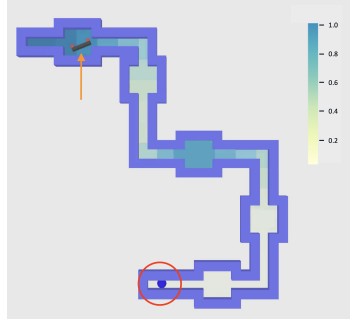 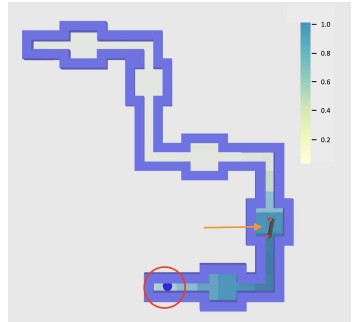

(a) Agent attending to its current location as well the next next chamber.

(b) Agent only attends to the next corner and chamber and not the past.

Figure 4: Mean attention of all heads of the $\mathbf{TR}^2$-GPT2 model. Orange arrow indicates where the agent is on the map, and the red circle indicates the goal to reach. Darker blue represents the most attention and lighter colors represent minimal attention.

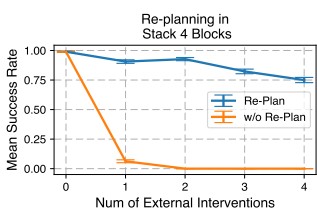 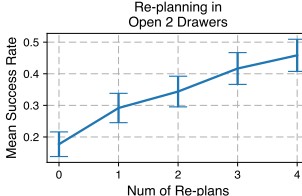 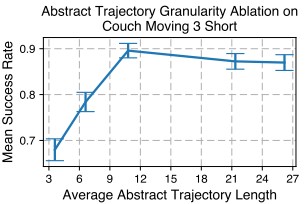

Figure 5: Evaluation of $\mathbf{TR}^2$-GPT2 on **(left)** stacking 4 blocks with varying amounts of external interventions, only re-planning once per intervention; **(middle)** opening two drawers with variable amounts of re-planning; **(right)** Couch moving with varying granularity of abstract trajectories.

the agent to successfully bridge the high- to low domain gap in Couch Moving. Moreover, the agent learns to pay attention mostly to locations up ahead and learns that the past parts are uninformative, despite being given the full abstract trajectory to process at each timestep. A video of full trajectory attention analysis can be found in the supplementary materials and our project page .

### 4.6 EFFECTS OF RE-PLANNING

One property of our approach is the feasibility of re-generating the abstract trajectory at test time, and we refer to this as *re-planning*. It enables us to introduce explicit error-corrective behavior via the high-level agent in long-horizon tasks.

Our results in Fig. 5 (left and middle) show that re-planning enables handling unforeseen interventions in Block Stacking and mistakes in Open Drawer successfully. In Block Stacking, we add external interventions where we randomly move blocks off the tower and allow the agent to re-plan just once per intervention. As the number of external interventions increases, the success rate does not lower as significantly compared to not re-planning since the re-generated abstract trajectories guide the low-level agent to pick up misplaced blocks. In Open Drawer, the robot arm must open two drawers and often will close the one it opened by accident. With re-planning, the high-level agent provides a corrective abstract trajectory to guide the low-level agent to re-open the closed drawer and succeed.

### 5 CONCLUSION

We have introduced the Trajectory Translation ($\mathbf{TR}^2$) framework that seeks to train low-level policies by translating an abstract trajectory into executable actions. As a result we can easily decouple plan generation and plan execution, allowing the low-level agent to simply focus on low-level control to follow an abstract trajectory. This allows our method to generalize to unseen long-horizon unseen tasks. We further can utilize re-planning via the high-level agent to easily improve success rate to handle situations when mistakes or external intervention occurs.

## 6 Reproducibility Statement

We have uploaded our anonymized code to a GitHub repo: `https://github.com/abstract-to-executable/code`. For details on running the training and evaluation code, see the README. For exact hyper-parameter settings check Appendix G.

## 7 Ethics Statement

A part of our work utilizes collected data in the form of abstract trajectories that will be released for other researchers to use. We want to reaffirm that this data is completely generated using code with no humans involved in physically generating data. Moreover, the generated data only encodes information about solving a few manipulation and navigation tasks.

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

**Appendix Table of Contents**

# A   REAL ROBOT EXPERIMENTS OF BLOCK STACKING

In this section we describe how the Block Stacking environment was constructed in the real world and how the real world experiments were conducted. Videos of the results can be found here: `https://sites.google.com/view/abstract-to-executable/real-world-videos`

## A.1   REAL WORLD ENVIRONMENT

The real world environment consists of a robot arm with a parallel gripper, an RGBD camera, and a flat table of which blocks can be placed.

**Robot Arm**   We use a UFactory xArm 7 robot with a xArm gripper as this was the closest arm matching that of the Panda arm and gripper used in simulation that was available to us. The arm and gripper is controlled via 5D position control of the end-effector (3) and the gripper (2).

**RGBD Camera**   We use the Intel RealSense sensor to capture RGBD data from the scene.

**Blocks**   We use 2-inch / 5.08 cm width wood blocks (which are built by taping eight 1-inch blocks) for block stacking. Each of the blocks also have a texture taped onto it made from paper cutouts using assets from the game Minecraft. Note that in simulation we actually use 4 cm width blocks, and to adjust for this we scale all positions by 5.08/4.00 accordingly when transferring to real.

## A.2   EXECUTION

In Fig. 6, we show different kinds of block configurations $\mathbf{TR}^2$-GPT2 tackles. On average each block will require around 40 to 60 actions and in our experiments we are able to stack up to 26 blocks in novel configurations requiring up to around 1600 actions. In order to perform Block Stacking in the real world setting, we conduct the following process

1. Detect the position of isolated blocks not at goal positions. Directly use the detected position to place the block in the simulated environment to create an initial state. See Sec. D for more details on how we estimate block positions in both real world and simulation.

2. Run the high-level agent and generate an abstract trajectory to pick up the block and place it in a goal position

3. In the simulated environment, run the trajectory translation model on the generated abstract trajectory and initial state to generate a executable trajectory.

4. Run the executable trajectory in real world by setting the position of the xArm based on the 3D position of the end-effector and 2D position of the grippers in the executable trajectory states.

As we simply wish to show feasibility of sim-to-real transfer, in practice we assume that in the real world the past placed blocks were successful, and we only care about estimating the position of the new block given to the robot arm to be placed. Note that usually the position of all the blocks would be given to the high-level agent for planning purposes in order to generate an abstract trajectory. In our experiments we're not concerned with estimating the position of every block although that would be possible with a more sophisticated vision pipeline. Moreover, the low-level policy only needs to observe the position of the block the high-level agent manipulates, so our simplification is more than feasible.

## A.3   FAILURE MODES

There are a few failure modes that arise when transferring from simulation to the real world. We detail them in order of general frequency.

1. The biggest problem was blocks bouncing and rotating when dropped onto another block or the table. In simulation, the block is released nearly perfectly from both grippers on the end-effector, resulting in minimal external rotation forces causing it to rotate and thus,

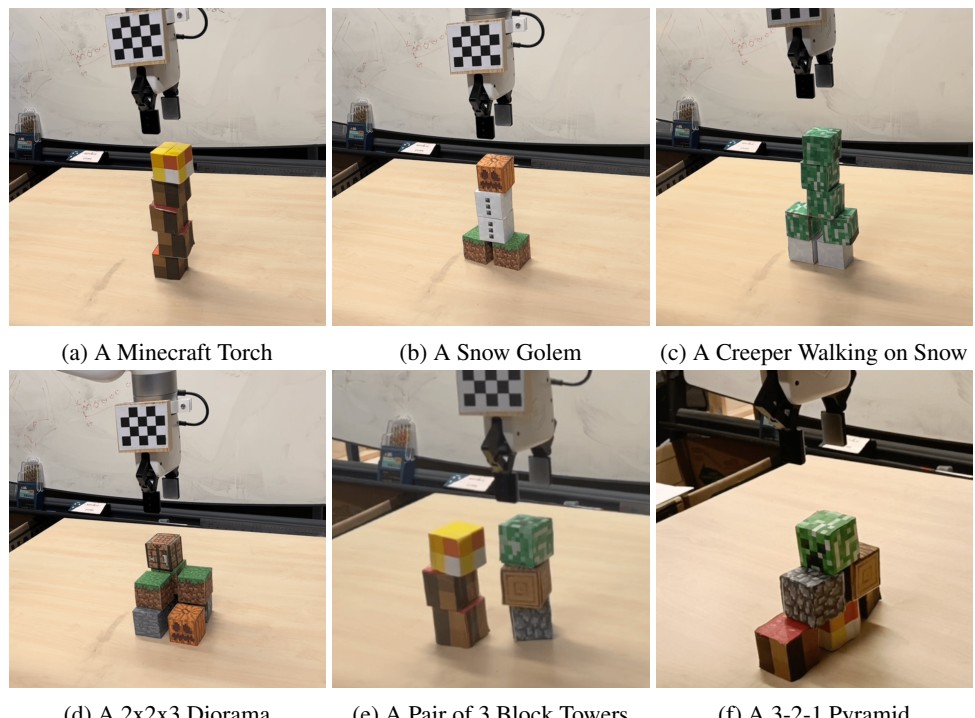

(a) A Minecraft Torch      (b) A Snow Golem      (c) A Creeper Walking on Snow

(d) A 2x2x3 Diorama      (e) A Pair of 3 Block Towers      (f) A 3-2-1 Pyramid

(g) A 28-block Castle. Note that the robot failed to stack two blocks properly towards the back left. In the end 26 out of 28 blocks were stacked

Figure 6: Example real-world **test** configurations built using the **TR**$^2$-GPT2 policy, which is trained on stacking a single block on the top of another one or two blocks in simulation. Test configurations vary in height of up to 5 blocks as well as using up to 26 blocks in trajectories that take up to 1600 steps to solve. Complexity varies from balancing blocks on one or multiple blocks to packing blocks closely together. Videos of real-world block stacking can be found in our supplemental materials.

the blocks land perfectly. However, in the real world this is not always possible as the blocks we used would stick a little to the xArm gripper, causing one side of the block to be released at a different time to the other side. This leads to imprecision and bouncing in block placement as well as unwanted rotation. We partially address this issue by engineering abstract trajectories to release blocks from a lower height so the model will also try and release blocks from a lower height, minimizing imprecision caused by rotation and bounces. We further mitigate this issue in some real world experiments by discretizing along the z-axis to constrain the robot arm to dropping the blocks at certain heights.

2. Another issue is the supported range of the learned policy. Since in training the learned policy only trains to pick and place single blocks in a predefined region, it can only generalize so far before having more and more error. In the real world experiments we didn't draw explicit boundaries marking what regions were within what the model had seen in training and what regions were not. Thus, when placing blocks in the real world sometimes we place them too far away and the model fails to pick it up. This is something that could be addressed by diversifying the training dataset further to include a wider spawn range of blocks.

3. Lastly, while this isn't a failure mode specific to the real world, it may appear at first glance that there is a discrepancy between simulation success and real world success. In simulation we can stack 6-block tall towers albeit with low success rates whereas in the real world we are able to stack a large castle configuration with over 3x the number of blocks. This is the result of the common failure mode where stacking higher towers is more difficult as it requires more and more generalization in vertical stacking since in training, agents only ever stack a single block up to a height of 3 blocks. Table 1 shows that stacking tower of height 4 has very high success rate, and since the castle configuration has only at most height 4 blocks, it is more than feasible to stack successfully.

## B    SPECIFICATIONS OF ALL ENVIRONMENTS IN MAIN PAPER

This section dives into specific details for each environment we test on, including the heuristic used for abstract trajectory generation, the exact details of what is in the high- and low- level state spaces, as well as a description of each task. We further show more examples of specific details that were left out of the main manuscript such as figures depicting how couch moving works.

### B.1    BOX PUSHER

**Environment configuration**    the low-level agent receives a 6 dimensional observation consisting of the 2D positions of the agent, the green target box, and the blue goal location. The low-level agent's action space is 2 dimensional and is simply a xy delta position vector. The high-level space has 4 dimensional high-level states containing the 2D position of the high-level pointmass and the 2D position of the target box. The high-level agent further can magically grasp the target box.

The general task is to control a box shaped agent to push a target green box as shown in Fig 1 to a blue goal location. The training task's variation does not include any obstacles, whereas in test time the task includes red obstacles that only the high-level agent can see, depicted in Fig. 3. Without guidance from a higher-level agent through an abstract trajectory, it is difficult to avoid the obstacles since they aren't in the observations. Thus, the test task requires careful following of an abstract trajectory in order to push the green target box to the goal.

Success is defined as when the target green box reaches within an $\epsilon$ distance of the blue goal location. $\epsilon$ is equal to the width of the target box.

**High- to low-level domain gap**    Under this setup, the high-level agent can attach itself to the green target box with magical grasping, and directly drag and pull the box around. The low-level agent however does not have magical grasp and can only push the green target box. In the example in Fig. 1 we show how the low-level agent must go behind the green target box, deviating from the high-level plan, in order to push it to the left whereas the high-level agent can simply move to the green target box and immediately start moving to the left.

**Abstract Trajectory Generation**    The heuristic to generate the trajectory is as follows

1. Move the agent along the x or y axis until the x or y position difference to the target box is below a threshold $\epsilon$. Then do the same for the other axis.

2. Once near enough to the target box, enable grasping and the target box will follow the agents every subsequent movement.

3. Move the agent along the x or y axis again until the x or y position difference between the target box and the goal location is below a threshold $\epsilon$. Then do the same for the other axis.

In total, there are 4 random variations of the abstract trajectories for any given starting high-level state, depending on which direction the agent approaches the target box first (2), and which direction the agent moves the target box to the goal location first (2).

## B.2    COUCH MOVING

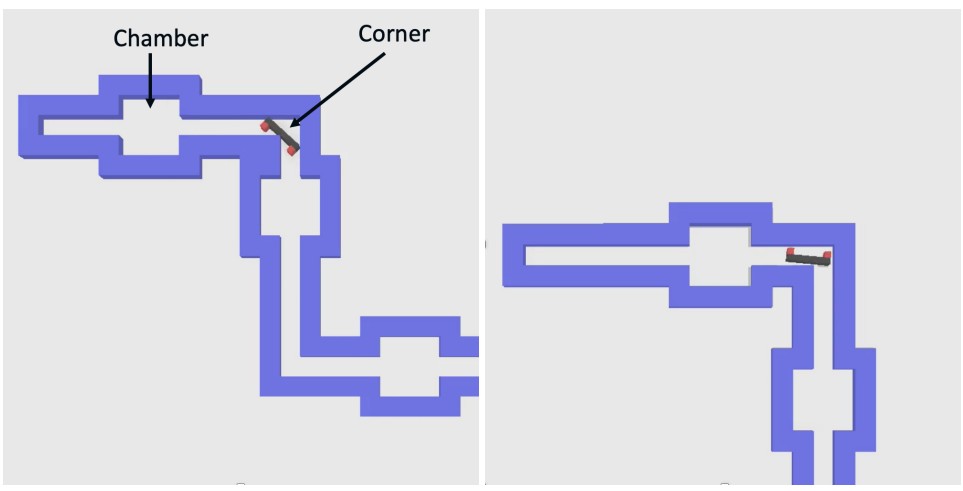

(a) Successful rotation of couch allows it to pass through a corner

(b) Unsuccessful rotation of couch prevents it from passing through a corner

Figure 7: 7a shows how a properly oriented couch can easily go through a corner. 7b shows that on the other hand the wrong orientation can prevent the couch from moving through the corner

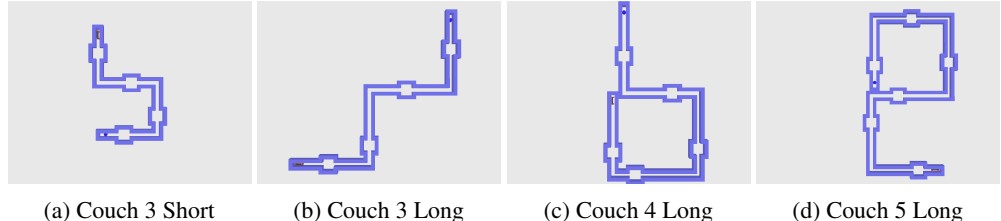

(a) Couch 3 Short        (b) Couch 3 Long        (c) Couch 4 Long        (d) Couch 5 Long

Figure 8: Various configurations of the Couch Moving environment. (a) is the train configuration. (b), (c), (d) are test configurations.

**Environment configuration**    The low-level, couch shaped, agent receives a 9D vector representing a local 3x3 patch of its surroundings, indicating empty space (0) and walls (1). Additionally it also receives a 2D vector representing which direction along the maze moves forward to the final goal, and it's own 2D position. The low-level agent has a 3-dimensional action space consisting of 2D force and 1D torque. The high-level space contains 2D dimensional states containing only the 2D position of the high-level agent pointmass.

The task is to control a couch shaped agent and move it through a map composed of chambers, corridors, and corners, which are marked in Fig. 7a. Success is defined as when the agent reaches within an $\epsilon$ distance of a blue target circle marked on the map.

The environment nomenclature is follows the structure Couch Moving [short/long] $n$. Long has corrdiors around 1.5x longer than the short variation, and $n$ is the number of corners in the maze. The training environment is Couch Moving Short 3 and the test tasks are all long variations with more corners. Fig. 8 shows visually these different configurations.

**High- to low-level domain gap**    In couch moving, the high-level agent and low-level agent rather have a large domain gap due to different morphologies. The high-level agent is a pointmass that can go through the entire map with no issues. However, the low-level agent is a couch shaped agent and cannot fit through corners unless oriented correctly as shown in Fig. 7. Furthermore, the low-level agent can only access a local patch of the map layout, while the high-level agent has access to the whole map. This requires the low-level agent to learn to attend to different parts of the abstract trajectory in order to determine when to rotate in chambers, and we investigate how **TR**$^2$-GPT2 bridges this gap via attention in Sec. 4.5

**Abstract trajectory generation**    The high-level agent is a point mass that can freely move around within the map walls. The high-level states consist just the absolute 2D position of the high-level agent. As a result the abstract trajectory is simply the map itself represented as a sequence of 2D positions.

### B.3 Block Stacking

**Environment configuration**    The low-level robot receives 32 dimensional observations from the environment, consisting of the robot arms joint position, joint velocity, the target block's pose, and the end effector's pose. The arm is controlled via a 4 dimensional delta position controller on the end effector. The high-level space consists of 6 dimensional vectors containing the 3D position of the high-level agent pointmass and the 3D position of the block to stack. The high-level agent also has magical grasping.

The training task is visualized in Fig. 9, and consists of a single pick and place of a single block up to a height of 3 blocks. The location of already placed blocks and the target block to be moved varies within a constrained region reachable by the robot arm. The robot arm also has a small amount of randomness in where it initializes.

The test tasks shown in our results in Table 1 are visualized in Fig. 10 and vary in complexity of number of blocks used to stack towers and pyramids in addition to a wider region in which blocks are to be placed. The test tasks are designed to test long horizon task generalization by stacking many blocks in farther locations in succession without fail. Additionally, the stacked configurations are different than what's seen in training time.

We additionally run real-world tests on a wider variety of configurations detailed in Sec. A.

Success is defined as placing every spawned block in a goal position and the robot gripper moving a minimum $\epsilon$ distance away from any of the stacked blocks. In our environment $\epsilon$ is equal to 2 times the width of the blocks.

**High- to low-level domain gap**    The high-level agent is a pointmass represented by a 3D coordinate position and can magically grasp blocks. However, the low-level agent contains information about the entire robot propioception as well as the pose of the end effector. Moreover, the low-level agent has complex dynamics as it must physically grasp and release blocks precisely.

**Abstract trajectory generation**    The high-level agent returns a trajectory consisting of 10-dimensional high-level states. This includes the 3D positions on the $x, y$, and $z$ axes of the high-level pointmass agent and the target block to stack. The following details the abstract trajectory generation heuristic

1. Move the high level agent to a random safe height. Then the high-level agent will elevate to another random height above the target block as it moves toward the target block.

2. The agent then moves down and approaches the target block. The high-level agent then turns magical grasp on and the target block will now follow the high-level agent's every subsequent movement.

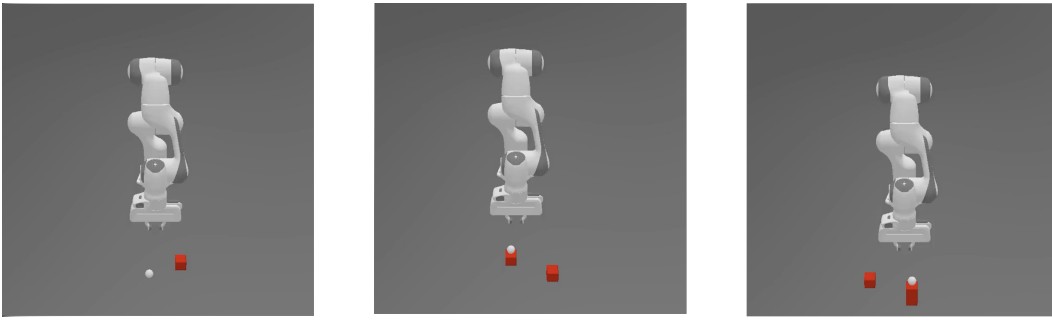

Figure 9: Examples of training time tasks. The low-level agent must follow an abstract trajectory and place a block at the designated white goal position

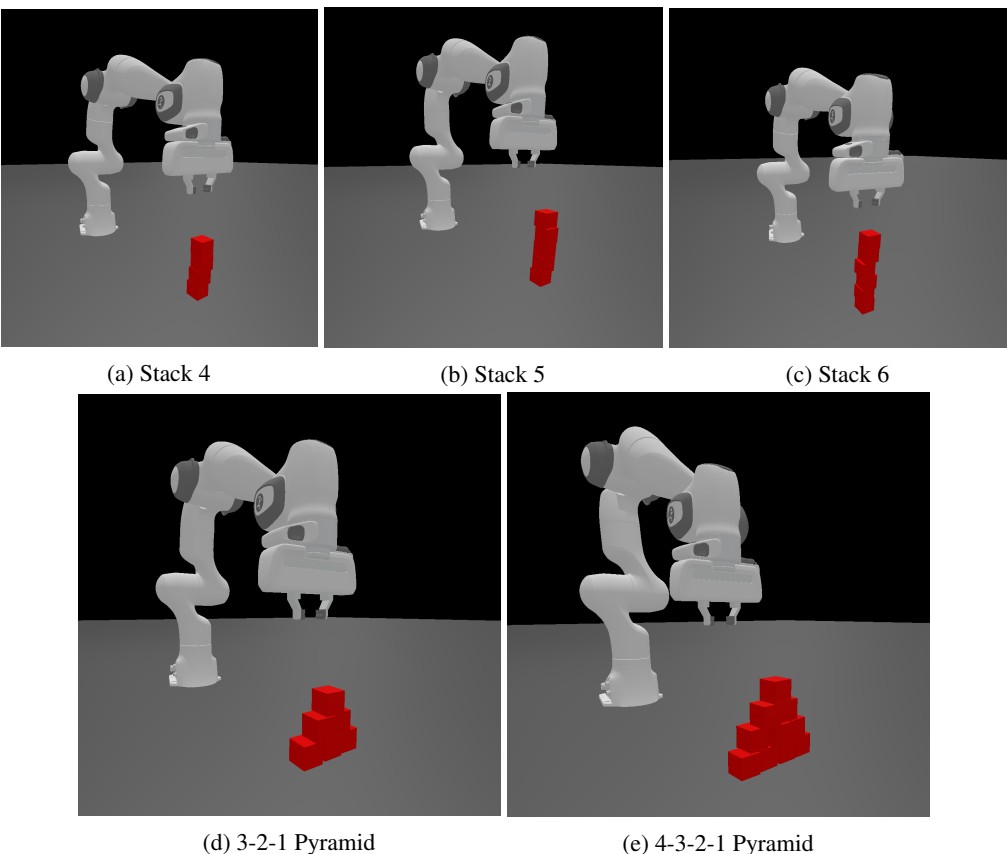

| (a) Stack 4 | (b) Stack 5 | (c) Stack 6 |
| --- | --- | --- |

| (d) 3-2-1 Pyramid | (e) 4-3-2-1 Pyramid |
| --- | --- |

Figure 10: The final completed builds of various test tasks such as stacking towers with 6 blocks or building 10 block pyramids with 4 layers.

3. The agent then goes up to a random safe height, moves to the top of goal location, and finally lowers down the block to a random height above the goal location.

4. The high-level agent stops magically grasping the target block and the target block will no longer move and stay where it was last placed.

5. Lastly, the high-level agent will move up to a random safe height, then move back to the location where it started.

There are various random variations in the abstract trajectory, these include the various random heights the agent moves up to before/after picking or placing, as well as when the agent releases the target block it is magically grasping. These variations can be learned by the **TR$^2$-GPT2** model and allows a user more fine-grained control over how exactly the robot arm picks up blocks by engineering the abstract trajectory.

### B.4    OPEN DRAWER

**Environment configuration**    The low-level robot receives 39 dimensional observations, which contains its joint position, joint velocity, pose of its end-effector and target handle's axis-aligned bounding box. Its actions are 13 dimensional and are all realized through a joint velocity controller. The high-level space consists of 9 dimensional states composed of the 3D position of the high-level agent pointmass and a 6-dimensional bounding box of the handle. The high-level agent can further magically grasp the handle and pull the drawer out with ease.

To produce the axis-aligned bounding boxes, we use the environment's point cloud data captured by a fixed camera. The handle's points in the point cloud are filtered out through a built-in segmentation map provided by SAPIEN, then these points are used to calculate an axis-aligned bounding box. This vision pipeline is expanded upon in Section D.2.

The training task is to open a single drawer on cabinets in the training set. One test task to test object generalization is to open a single drawer on unseen cabinets in the test set. The final test task is to open two drawers on cabinets with at least two openable drawers, which is difficult to solve as it is easy to accidentally close a drawer while opening another.

Success is defined as opening all targeted drawers at least 90% of the way and the drawers maintaining a velocity less than a threshold.

**High- to low-level domain gap**    The high-level agent is a pointmass represented by a 3D position and can magically grasp and move the target drawer open. The low-level agent however is a 13-DoF robot and has complex dynamics and must physically grasp and control the drawer handle and pull it open.

**Abstract trajectory generation**    The high-level state is 9 dimensional, containing the high-level agent's 3D positions and the bounding box of the target handle. Note that in high-level states the bounding box is extracted first from the given starting low-level state, and is then magically moved through space via translation only. We do not recompute the bounding box of the target handle or capture new point cloud data as the high-level agent generates the abstract trajectory.

The details of the abstract trajectory generation implementation is described below:

1. The high-level agent will first move up to a height equal to the mean height of the captured bounding box for target handle given by the starting low-level state. Then, the high-level agent will approach the handle until it is close enough.

2. The high-level agent will turn on magical grasping, meaning every subsequent movement of the high-level agent will also move the axis-aligned bounding box captured from the initial state.

3. The high-level agent will begin moving in the direction that opens the drawer until it is opened completely

4. The high-level agent finally releases the magical grasp and moves to a random new location away from the handle and cabinet.

The only variation in the abstract trajectories are where the high-level agent decides to move to after opening the cabinet.

## B.5   OBSTACLE PUSH (SILO)

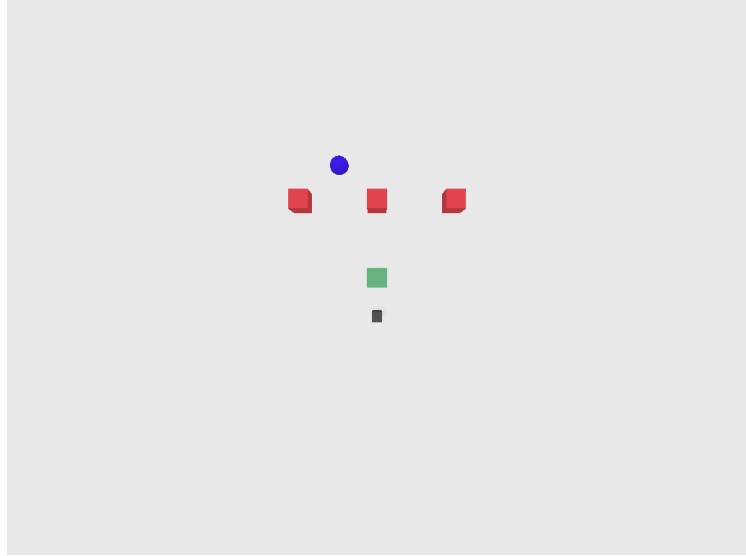

Figure 11: Example of the recreated Obstacle Push environment from SILO (Lee et al., 2019). This is adapted from our Box Pusher environment, with the black stick now representing the closed gripper used in SILO, the red boxes representing obstacles, and the blue target representing the goal location.

**Environment configuration**   The task is to control a stick-like agent to push the green box to the target location as depicted in Fig. 11. This environment was recreated based off the original paper by Lee et al. (2019) as their environment code could not be found at the time of writing.

**High- to low-level domain gap**   The high-level agent is simply a point mass with magical grasp and the magical ability to move the target green box through obstacles. The low-level agent on the other hand must deal with the obstacles and manipulate the box to move around the obstacle and to the goal.

**Abstract trajectory generation**   The abstract trajectory generated here is meant to mimic as closely as possible the demonstrations generated in SILO for this environment. The details of the abstract trajectory generation implementation is described below:

1. The high-level agent will first move forward to the green box and magically grasps it.

2. The high-level agent then drags the grasped box along with it as it moves straight upwards until it goes through the middle red obstacle and reaches the same x-axis as the blue target.

3. The high-level agent moves straight to the blue goal until the grasped green block is on the blue goal.

## B.6   PICK AND PLACE (SILO)

**Environment configuration**   The task is to control the robot arm to pick up the red block and bring it to the goal 12. The action-space is the same exact 4-DOF action space used by SILO and the observation space is mostly the same barring some differences between Panda and Sawyer arms. This environment was recreated based off the original paper by Lee et al. (2019) as their environment code could not be found at the time of writing.

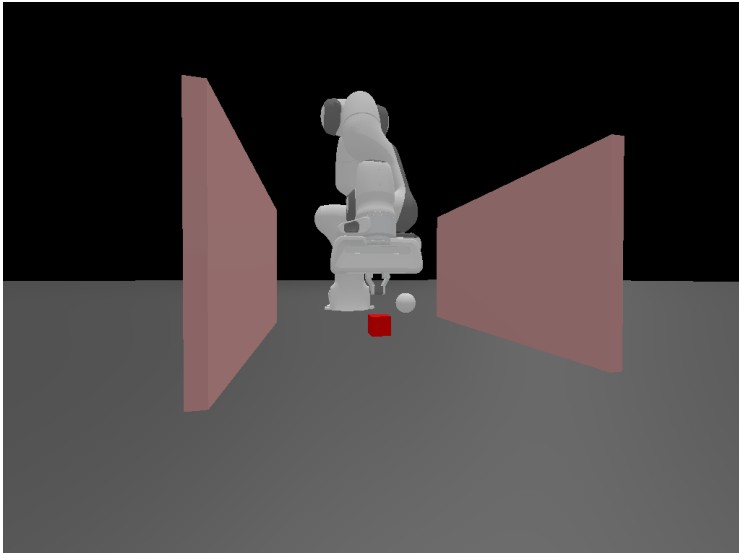

Figure 12: Example of the recreated Pick and Place environment from SILO (Lee et al., 2019). This is adapted from our Block Stacking environment, but instead we have the panda arm and gripper instead of the Sawyer one. The goal location of the block is the white sphere

**High- to low-level domain gap**    The high-level agent is simply a point mass with magical grasp and the magical ability to move the red block through the two walls. The low-level agent on the other hand must deal with the two walls and manipulate the box to mimic the high-level agent as much as possible to within feasibility as the walls prevent the low-level agent from fully following the high-level agent.

**Abstract trajectory generation**    The abstract trajectory generated is based on the details supplied by SILO. The details of the abstract trajectory generation implementation is described below:

1. The high-level agent will first to the red block and magically grasp it.

2. The high-level agent then moves the grasped box along with it through two milestones in a bit of a curve. One milestone is either to the left or the right of the initial starting point, the 2nd milestone is the goal.

## C    TRAINING

This section explains how we train our trajectory translation models as well as detailing the reward functions used.

### C.1    ONLINE TRAINING

We use PPO (Schulman et al., 2017) as our training algorithm. Both the actor and critic are initialized with the same model but separate weights. For additional architecture details see Sec. E. In all experiments, online training hyperparameters are mostly kept the exact same, see Sec. G for specific hyperparameters used. Moreover, all results reported are averaged over three seeds.

In particular, for the Block Stacking environment, after the initial online training, only the **TR**$^2$-GPT2 model had any substantial success rates. We further train the **TR**$^2$-GPT2 model in a second round where we simply turn gradient accumulation on and continue training, reducing the number of gradient updates in each epoch to just 3. This helped improve the maximum success rate the models were able to attain. Lastly, in practice the dissimilarity function $d$ associated with the mapping function will weigh data relevant to the agent by 0.1 and data relevant to all other objects by 0.9.

### C.2 ADDITIONAL DISCUSSION ON THE TRAJECTORY FOLLOWING REWARD

Here we discuss and show an example regarding the limitation of the trajectory following reward when the abstract trajectory has repeated/similar high-level states.

An example of this is in periodic tasks such as repeated pick and place between two locations. Using the $TR^2$ framework, a practical way to overcome this issue is to chunk the abstract trajectory into parts where there is no periodicity. For repeated pick and place for example, each chunk would be a smaller abstract trajectory corresponding with each pick and place. See this this video for an successful example of applying this strategy to the task of repeatedly pick and placing a block between two locations three times.

### C.3 TRAJECTORY FOLLOWING RETURN TRACE

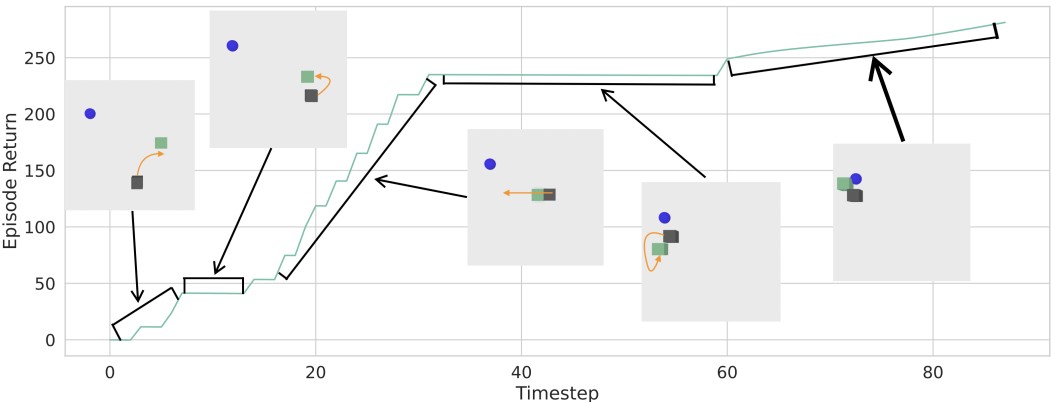

Figure 13: Episode return trace for a single episode. The chart only shows trajectory following rewards and does not include the scaled down task reward. Orange arrows indicate direction of future movement of the black agent

We show how the trajectory following reward (Eq. 1) works visually using Box Pusher as an example through the return trace in Fig. 13. Step by step, each snapshot of the Box Pusher environment in Fig. 13 is explained as follows:

1. The increase in return in the first few time steps is attributed to the low-level agent following the high-level agent to move towards the target green box.

2. Then the following plateau is when the low-level agent must go behind the target green box before pushing it, showing how in this scenario, we do not give more reward as the low-level agent is not matching new high-level states.

3. Once the agent starts pushing the target green box, it begins to match new high-level states as the target green box is moving along the path demonstrated in the abstract trajectory, leading to more reward.

4. In the next plateau in return, we see that the low-level agent tries to push the target green box up to match the abstract trajectory, but requires around 30 steps to move behind the target green box, in which it is not matching new high-level states.

5. Finally, the low-level agent pushes the target green box to the blue goal location and gains constant reward for keeping the target green box at the blue goal location.

### C.4 TASK REWARD

Each task comes with a basic task reward to define the task. And we use the task reward as an auxiliary supervision signal in addition to the trajectory following reward. In this section, we present the task rewards.

### C.4.1 Box Pusher

We define two distances, $d_{a,b} = ||p_a - p_b||$, $d_{b,g} = ||p_b - p_g||$ where $p_a = $ 2D position of agent, $p_b = $ 2D position of target box, and $p_g = $ 2D position of the goal location. The task reward at each timestep is then equal to $-(0.1d_{a,b} + 0.9d_{b,g})$.

In the trajectory following setting, the task reward is scaled down by a factor of $0.1$.

### C.4.2 Couch Moving

Whenever the 2D position of the agent $p_a$ is in a chamber, we give 0 reward.

Whenever $p_a$ is more than $\epsilon$ distance away from the center of any chamber, we give reward 0 if it is oriented correctly for the upcoming corner as shown in Fig. 7a, and reward $-1$ if it is not oriented correctly as shown in Fig. 7b. In the trajectory following setting, the task reward is scaled down by a factor of $0.5$.

Note that this reward function is not meant to be well defined, and as a result goal conditioned policies will have a lot of trouble having any meaningful success. The task reward here is to simply aid the exploration process for models in the $\mathbf{TR}^2$ framework when trained online.

### C.4.3 Block Stacking

The low-level agent is tasked with picking up a block, stacking it at the goal location, then returning back to its initial location. Thus our success metric is:

$$\text{success} = \begin{cases} 1 & \text{if} \quad ||p_{init} - p_a||^2 < \epsilon_1 \quad \text{and} \quad ||p_a - p_g||^2 > \epsilon_2 \\ & \text{and } \textbf{not grasping} \text{ and } \textbf{close enough} \\ 0 & \text{otherwise} \end{cases} \tag{2}$$

The success metric relies on four conditions: firstly the low-level agent is asked to returned to its initial location; secondly, the agent is required to move away from the goal position greater than a certain distance; thirdly, the agent must not be grasping the block; and lastly, the target block needs to be a close enough to the goal location. A way to check if the agent is still grasping a block is implemented as:

$$\text{grasping} = \begin{cases} \text{True} & \text{If for left and right finger, angles between} \\ & \text{impulses and open directions are smaller than threshold} \\ \text{False} & \text{Otherwise} \end{cases} \tag{3}$$

To assist the agent in task completion, we employ a four-stage task reward to encourage it to approach, grasp, and transport the block to the goal position, and return to initial location. To prevent RL agents from remaining in an intermediate stage indefinitely, we ensure that the rewards in each successive step are strictly greater than those in the current stage.

$$R^{Task} = \begin{cases} 1 - \tanh(||p_b - p_a||^2) & \textbf{not grasping} \text{ and } \textbf{not close enough} \\ 2.25 - \tanh(||p_b - p_a||^2) - \tanh(||p_g - p_b||^2) & \textbf{grasping} \text{ and } \textbf{not close enough} \\ 3.5 & \textbf{grasping} \text{ and } \textbf{close enough} \\ 14.5 - \tanh(||p_{init} - p_a||^2) & \textbf{not grasping} \text{ and } \textbf{close enough} \end{cases} \tag{4}$$

where $p_b, p_g, p_a$ represents the target block's position, the goal position, and the agent's position respectively. At stage one, the agent is not grasping the target block and the target block is not close enough to the goal location, so the agent receives a reward encouraging it to approach the target block. A constant reward will be given to the agent if it is grasping the block, and upon grasping, the reward enters the second stage. At stage two, the agent is grasping the target block while the target block is not at the goal location, thus an additional training signal will be added to the reward to encourage the agent to take the block to the goal location. When the target block gets close enough to the goal

location, an extra reward will be given to the agent so that it stays close to the goal location. During the last stage, the agent has released the target block at the correctly location, and a reward will be given to the agent to encourage it to go back to initial location.

In the trajectory following setting, the task reward is scaled down by a factor of $0.2$.

### C.4.4   OPEN DRAWER

Since this task is adpated from the OpenCabinetDrawer task in ManiSkill benchmark(Mu et al., 2021), we just use the original task reward from the ManiSkill benchmark, and we briefly explain the reward below.

The Open Drawer environment divides the reward into three stages. In the first stage, the agent is rewarded for being close to the target drawer's handle. To promote contact with the target link, the negative value of Euclidean distance between the handle and the gripper is added into the reward. When the distance between the gripper and the target link is smaller than a threshold, the agent proceeds to the second stage. The agent receives a reward based on the opening angle of the door or the opening distance of the drawer at this stage. When the agent opens the door or drawer enough, the last period starts. At the very last stage, the agent gets a negative reward depending on the speed to encourage itself to remain static.

In the trajectory following setting, the task reward is scaled down by a factor of $0.02$. Note that the task reward weight is much smaller than in other environments since the overall return of Open Drawer's task reward is much larger in magnitude than in other environments.

### C.4.5   OBSTACLE PUSH (SILO)

The same task reward for Box Pusher is used here as well.

### C.4.6   PICK AND PLACE (SILO)

The same task reward for Block Stacking is used here as well.

## D   VISUAL INPUT PROCESSING

We use visual inputs in Block Stacking and Open Drawer, and this section explains how we process the visual inputs.

### D.1   BLOCK STACKING

In the Block Stacking environment, we need to estimate the 3D position of the blocks. In our code, our models in fact use the 3D position plus a 4D quaternion as input, however the quaternion rarely deviates from a fixed value, so we treat the models as simply only observing 3D positions.

In both simulation and the real world, we follow this general pipeline:

1. Capture an unstructured point cloud and use camera intrinsics and extrinsics to transform the point cloud into the world frame relative to the robot arm base.
2. Using hard-coded xyz boundaries filter out points outside of the working space where blocks are placed and stacked
3. Apply $k$-means clustering to segment the point cloud
4. Apply the Iterative Closest Point (ICP) algorithm to estimate block poses for each segmented point cloud. Note that we only keep the block position.

For the $k$-means step, we optimize $k$ by selecting $k$ that minimizes a scaled inertia score $i + \alpha k$. $i$ is the inertia, equal to the within-cluster sum-of-squares (Buitinck et al., 2013), and $\alpha$ is a cluster penalty factor.

The next two subsections on simulation and real world will provide additional details of how its done in those settings.

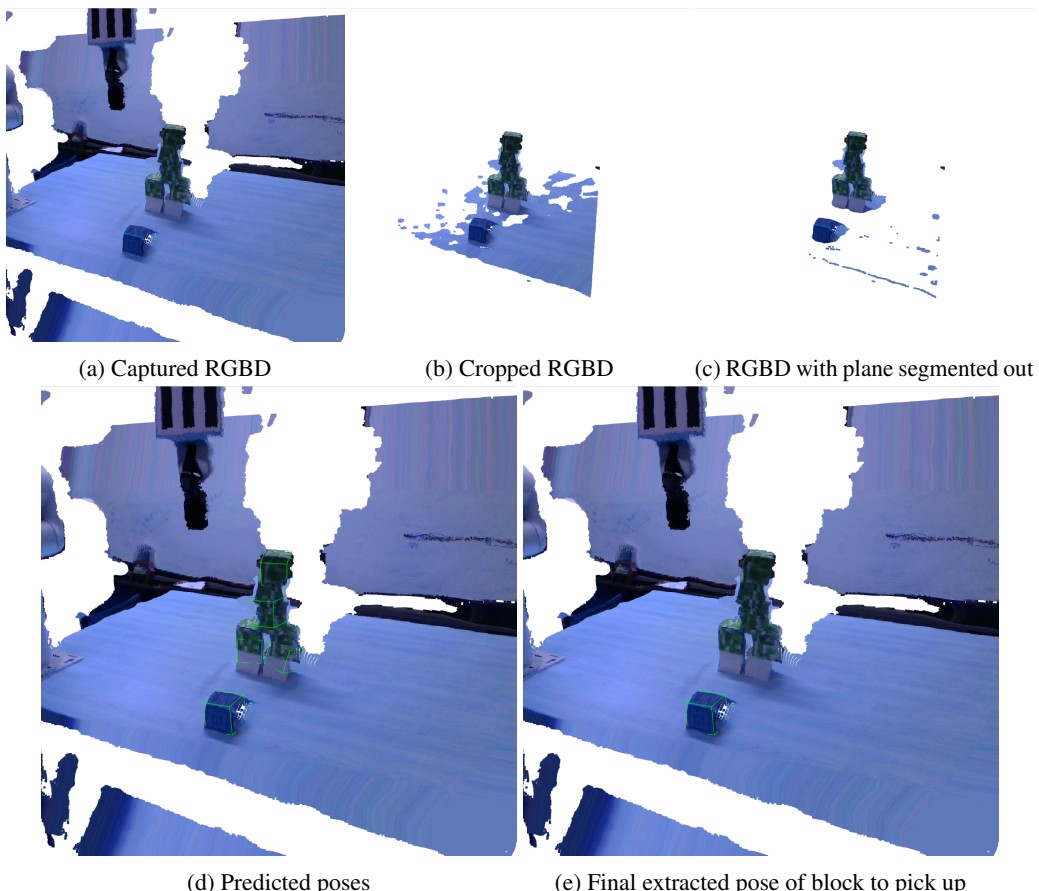

(a) Captured RGBD        (b) Cropped RGBD        (c) RGBD with plane segmented out

(d) Predicted poses        (e) Final extracted pose of block to pick up

Figure 14: Frame by frame process of how the pose of the block to manipulate is estimated in the real world.

### D.1.1 SIMULATION

All our results for Block Stacking in Table 1 are produced using estimated 3D block positions at every time step. Note that at training time, the policies use ground truth block positions in order to bypass the vision pipeline and speed up training.

In the SAPIEN simulator, we capture RGBD data as an unstructured point cloud and transform it into a world frame point cloud relative to the robot arm using the camera intrisics and extrinsics given by the simulator. In addition to the hard-coded xyz boundaries, we further remove the floor which is at $z = 0$. We further filter out points that are not red, leaving us with a point cloud in the world frame of just points of blocks in the scene.

For the k-means clustering for segmentation, we set the $\alpha$ parameter to $0.5$. Note that during evaluation in simulation, we only use the position of the block to be manipulated for the low-level agent, determined by the high-level agent.

### D.1.2 REAL WORLD

Using the Intel RealSense depth camera, we get RGBD data in the form of an unstructured point cloud. We are given the camera intrinsics by the camera, and we estimate the camera extrinsic via hand-eye calibration. Using the intrinsics and extrinsics, we transform the point cloud into the robot base's frame.

In addition to the hard-coded xyz boundaries that crops the RGBD into the points shown in Fig. 14b, we further remove most of the floor points by segmenting out the plane using RANSAC as shown in Figure 14c

For the k-means clustering for segmentation, we set the $\alpha$ parameter to $1.75$. After applying ICP, the resulting estimated block poses for each cluster is shown in Fig. 14d.

As our paper's focus is not on sim2real, in practice we simplify experiments by only using the position of the block closest to the camera and assume that is the block to be picked up and stacked to a desired goal position. The final desired pose is shown in Fig. 14e. More advanced vision pipelines can be used to better segment out individual blocks if necessary.

## D.2   OPEN DRAWER

As mentioned in section B, Open Drawer environments uses a point cloud representation to record its state which grants it the potential to be transferred to real experiments. These points are first filtered out by SAPIEN to include only points on the target drawer handle. The filtered points are then used to find the bounding box for target handle. Since the bounding box is aligned along axes, it is represented by a 6D vector as following:

$$\text{Bounding Box} := (x_{min}, y_{min}, z_{max}, x_{max}, y_{max}, z_{min})$$

where the minimum and the maximum among all points are recorded.

## E   TRAJECTORY TRANSLATOR IMPLEMENTATION DETAILS

This section describes the model architecture we use for $\textbf{TR}^2$ based models as well as some practical implementation details. When discussing the architecture in this section we will refer to Fig. 2.

**Abstract Trajectory Pre-processing**   In practice, we preprocess the abstract trajectory before training to constrain the distances between high-level states to a fixed value as well as to ensure the abstract trajectory is not too long while still being descriptive.

We preprocess abstract trajectories $\tau^H$ by interpolating between high-level states such that for a given high-level state $s_i^H$, the next high-level state $s_{i+1}^H$ is approximately $\epsilon$ distance away. If two adjacent high-level states from the original $\tau^H$ are closer than $\epsilon$ distance to each other, then we throw out the later high-level state. Higher $\epsilon$ means we reduce the length of the abstract trajectory while lower values increases the length.

$\epsilon$ can vary between environments and is tuned accordingly such that no two high-level states are too close and a low-level agent can't easily skip a high-level state by moving too fast. With the trajectory following reward, we observe that with lower $\epsilon$ values, $\textbf{TR}^2$-GPT2 often generates very jittery trajectories by moving back and forth in order to emulate moving slower and match every high-level state and so typically we will set $\epsilon$ higher.

Lastly, we will always retain the first and last high-level states.

**Abstract Trajectory Sub-sampling**   During training and evaluation, we sub-sample the abstract trajectory in order to improve inference speeds and training speeds. In particular, given abstract trajectory $\tau^H = (s_1^H, s_2^H, ..., s_n^H)$, we keep $s_n^H$, and interval sample the sequence $s_1^H, ..., s_{n-1}^H$ by keeping every $p-$th state $s_1^H, s_{p+1}^H, ....$ The sub-sampled abstract trajectory is then given to the low-level agent as part of the observation. Note that for trajectory following reward calculations, we still use the full un-sampled version of the abstract trajectory.

**Inputs**   The inputs at timestep $t$ to models in the $\textbf{TR}^2$ framework consist of the abstract trajectory $\tau^H = (s_1^H, s_2^H, ..., s_n^H)$ as well as the past executed low-level states $s_{t-k+1}^L, s_{t-k+2}^L, ..., s_t^L$. The abstract trajectory can also be viewed as a prompt, with the difference being that this prompt is always a part of the input sequence. The low-level states are generated in an auto-regressive manner via interaction with the environment, similar to the auto-regressive nature of the Decision Transformer (DT) (Chen et al., 2021).

The high-level states are passed through their own encoder, which can be a MLP, Convolutional Neural Network (Krizhevsky et al., 2017), PointNet (Qi et al., 2016) etc. depending on how you wish to process the data. In our work we only use the MLP as it is much faster to train with state based inputs. Similarly, the low-level states are passed through their own separate encoder as well.

The encoders output tokens $x_1, x_2, ..., x_{n+k}$ forming a sequence of length $n+k$. Positional encodings are removed and instead you can optionally add timestep based embeddings to the tokens related to the high-level states as Decision Transformer does.

The sequence is then fed through a model, which in our work is mainly the GPT2 Transformer but can easily be any other sequence processing model like LSTMs. The sequence processing model then outputs output tokens $z_1, z_2, ..., z_{n+k}$.

The $z_{n+k}$ token can be viewed as a contextual token that is then fed into the final MLP to guide the MLP in producing the final action that would make the agent follow the abstract trajectory.

Note that by feeding the sequence in the order presented above, unidirectional transformers like GPT2 enable the final output token $z_{n+k}$ to attend to all past executed states as well as the entire abstract trajectory, which is crucial for long horizon dependency modelling and is further examined in 4.5

## F ADDITIONAL RESULTS

We further benchmark our **TR$^2$**-GPT2 method on the X-Magical benchmark from (Zakka et al., 2022) as our method can also be seen as a sort of cross-embodiment learning method. We describe the environment setup below and results. Note that similar to Block Stacking, we utilize the extra fine-tuning stage for the Gripper embodiment.

### F.1 X-MAGICAL

#### F.1.1 ENVIRONMENT DETAILS

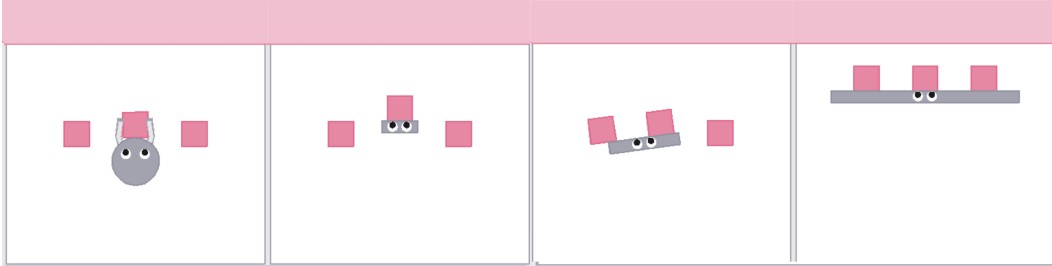

Figure 15: Examples of different low-level embodiments (from left to right: Gripper, Short-stick, Medium-stick, Long-stick), all solving the same task of pushing 3 boxes to the pink end-zone at the top

**Environment configuration** The task is to control an agent to push all three boxes into the end-zone as depicted in figure 15. We follow the same exact environment configurations as Zakka et al. (2022) and defer the reader to their paper for exact details. The original environment configuration is our low-level agent setup. The high-level space like our other environments abstracts all objects and the agent as simple 2D positions. Concretely, the high-level state is a 8 dimensional vector containing the 2D position of the pointmass and the three boxes.

**High- to low-level domain gap** The high-level agent is simply a point mass with magical grasp, allowing itself to easily move boxes to the end-zone one by one. Each low-level agent embodiment affects the way the low-level agent can feasibly push all three boxes to the end-zone. The gripper and short-stick embodiments are most similar to the high-level agent as they also move boxes to the end-zone one by one, although without the power of magical grasping. The medium-stick and long-stick embodiments are more different than the high-level agent in that they can move two and three boxes respectively at once and do not have magical grasp as well.

| Embodiment | $\mathbf{TR}^2$-GPT2 | XIRL |
|---|---|---|
| Short-stick | 79.0±3.6 | 81.4±3.4 |
| Medium-stick | 96.9±1.5 | 88.1±2.9 |
| Long-stick | 100±0 | 100±0 |
| Gripper | 63.3±4.3 | 72.3±4.0 |

Table 3: Results on X-Magical benchmark

**Abstract trajectory generation**   The details of the abstract trajectory generation implementation is described below:

1. The high-level agent will first move to a random box not in the end-zone and when within a $\epsilon$ distance, magically grasps onto the targeted box.

2. The high-level agent then drags the grasped box along with it as it moves straight to the end-zone.

3. The high-level agent releases the grasped box onto the end-zone and moves back out.

4. Repeat steps 1-3 until all boxes are in the end-zone.

### F.1.2   TASK REWARD FOR X-MAGICAL

A denser task reward is not defined for this environment in our experiments. We simply use the sparse success signal at the end of the episodes as the additional task reward and scale it down by a factor of 0.2.

### F.1.3   RESULTS

We achieve comparable results to XIRL on the X-Magical benchmark shown in Table 3. Note that we did not leverage access to a low-level dataset of (image observation-only) demonstrations, and instead utilize the generated abstract trajectories and the trajectory following reward to guide the agent to solving the task. Moreover, we like to point out that our method is conditioned on an abstract trajectory whereas XIRL trains goal conditioned policies.

## G   HYPERPARAMETERS

We detail the hyperparameters of our training processes. For all online training experiments we use the following common hyperparameters in Table 4. In subsequent tables 5, 6, 7, 8, 9, 10, we detail only the differences relative to the defaults.

| Hyperparameter | | Default | $\mathbf{TR}^2$-GPT2 | $\mathbf{TR}^2$-LSTM | SGC | GC |
|---|---|---|---|---|---|---|
| Optimizer | | Adam | - | - | - | - |
| Policy LR | | 3e-4 | - | - | - | - |
| Value Function LR | | 3e-4 | - | - | - | - |
| Rollout Batch Size | | 20000 | - | - | - | - |
| Training Batch Size | | 1024 | - | - | - | - |
| Epochs | | 2000 | - | - | - | - |
| Gradient Updates per Epoch | | 60 | - | - | - | - |
| Max Episode Length | | 200 | - | - | - | - |
| Number of Parallel Envs | | 20 | - | - | - | - |
| Initial Log Std Scale | | -0.5 | - | - | - | - |
| Target KL | | 0.15 | - | - | - | - |
| Clip Ratio | | 0.2 | - | - | - | - |
| GAE Lambda | | 0.95 | - | - | - | - |
| Discount Factor | | 0.99 | - | - | - | - |
| Reward Function | | Trajectory Following | - | - | - | Task |
| Trajectory Following Reward - $\beta$ | | 5 | - | - | - | N/A |
| Trajectory Following Reward - $w$ | | 30 | - | - | - | N/A |
| Max Abstract Trajectory Length | | 32 | - | - | N/A | N/A |
| Trajectory Sample Skip Steps ($p$) | | 2 | - | - | N/A | N/A |
| Stack Size ($k$) | | 2 | - | - | N/A | N/A |
| Timestep Embeddings | | True | - | - | N/A | N/A |
| Sequential Module Layer Dims | | [128,128,128,128] | - | - | N/A | N/A |
| Dropout | | 0.1 | - | - | 0 | 0 |
| State Embedding Activation | | ReLU | - | - | - | - |
| State Embedding Dims | | 32 | - | - | - | - |
| Actor MLP Activation | | Tanh | - | - | - | - |
| Actor and Critic MLP Layer Dims | | [128,128] | - | - | - | - |

Table 4: Default training hyperparameters during online training with PPO. The hyperparameters are split into three parts: PPO related, environment and reward function related, and model architecture related. Furthermore, the hyperparameters shown here are also the hyperparameters used for the Box Pusher experiments

| Hyperparameter | | Default | $\mathbf{TR}^2$-GPT2 | $\mathbf{TR}^2$-LSTM | SGC | GC |
|---|---|---|---|---|---|---|
| Max Episode Length | | 150 | - | - | - | - |
| Max Abstract Trajectory Length | | 50 | - | - | N/A | N/A |
| Trajectory Sample Skip Steps ($p$) | | 10 | - | - | N/A | N/A |
| Stack Size ($k$) | | 5 | - | - | N/A | N/A |
| Timestep Embeddings | | True | - | - | N/A | N/A |
| State Embedding Dims | | 64 | - | - | - | - |

Table 5: Couch Moving (Online) Training Hyperparameters

| Hyperparameter | | Default | $\mathbf{TR}^2$-GPT2 | $\mathbf{TR}^2$-LSTM | SGC | GC |
|---|---|---|---|---|---|---|
| Max Abstract Trajectory Length | | 55 | - | - | N/A | N/A |
| Trajectory Sample Skip Steps ($p$) | | 1 | - | - | N/A | N/A |
| Stack Size ($k$) | | 5 | - | - | N/A | N/A |
| Timestep Embeddings | | False | - | - | N/A | N/A |
| State Embedding Dims | | 64 | - | - | - | |

Table 6: Block Stacking Training Hyperparameters

| Hyperparameter | Default | $\mathbf{TR}^2$-GPT2 | $\mathbf{TR}^2$-LSTM | SGC | GC |
|---|---|---|---|---|---|
| Max Abstract Trajectory Length | 32 | - | - | N/A | N/A |
| Trajectory Sample Skip Steps ($p$) | 1 | - | - | N/A | N/A |
| Stack Size ($k$) | 5 | - | - | N/A | N/A |
| Timestep Embeddings | False | - | - | N/A | N/A |
| State Embedding Dims | 64 | - | - | - | - |

Table 7: Open Drawer Training Hyperparameters

| Hyperparameter | $\mathbf{TR}^2$-GPT2 |
|---|---|
| Max Abstract Trajectory Length | 48 |
| Trajectory Sample Skip Steps ($p$) | 2 |
| Stack Size ($k$) | 3 |
| Timestep Embeddings | False |
| State Embedding Dims | 32 |

Table 8: X-Magical Training Hyperparameters

| Hyperparameter | $\mathbf{TR}^2$-GPT2 |
|---|---|
| Rollout Batch Size | 10000 |
| Training Batch Size | 512 |
| Max Abstract Trajectory Length | 12 |
| Trajectory Sample Skip Steps ($p$) | 1 |
| Stack Size ($k$) | 2 |
| Timestep Embeddings | False |
| State Embedding Dims | 32 |

Table 9: Obstacle Push (SILO) Training Hyperparameters

| Hyperparameter | $\mathbf{TR}^2$-GPT2 |
|---|---|
| Max Abstract Trajectory Length | 30 |
| Trajectory Sample Skip Steps ($p$) | 0 |
| Stack Size ($k$) | 5 |
| State Embedding Dims | 64 |

Table 10: Pick and Place (SILO) Training Hyperparameters

# H  Learning Curves

This section shows figures for all the training success rates during online training. Note that Open Drawer was run at a different time and recorded the logs to Weights and Biases instead so the graphs are styled differently. All error areas shown represent standard deviation across seeds.

For clarification, some training success rates are lower than the reported results in Table 1. This is because the curves shown here are training success rates which use the Gaussian policy in PPO. Table 1 uses the mean of the Gaussian during evaluation so success rate is generally higher.

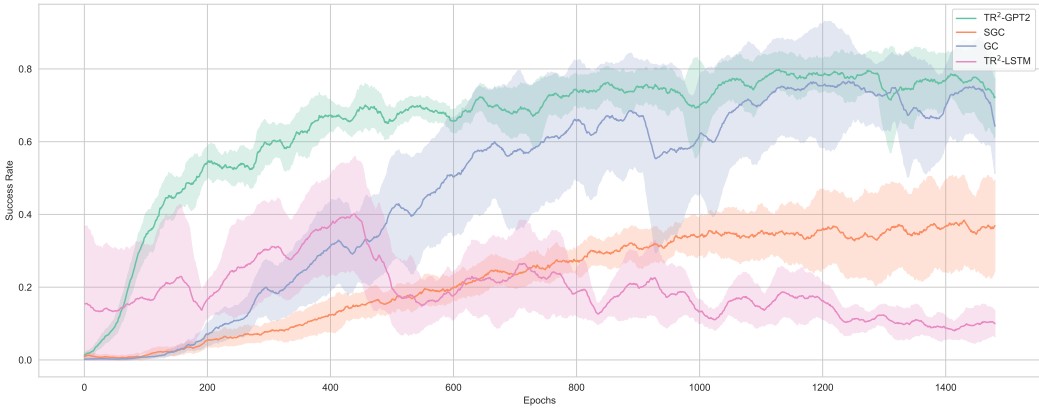

Figure 16: Box Pusher Training Success Rates

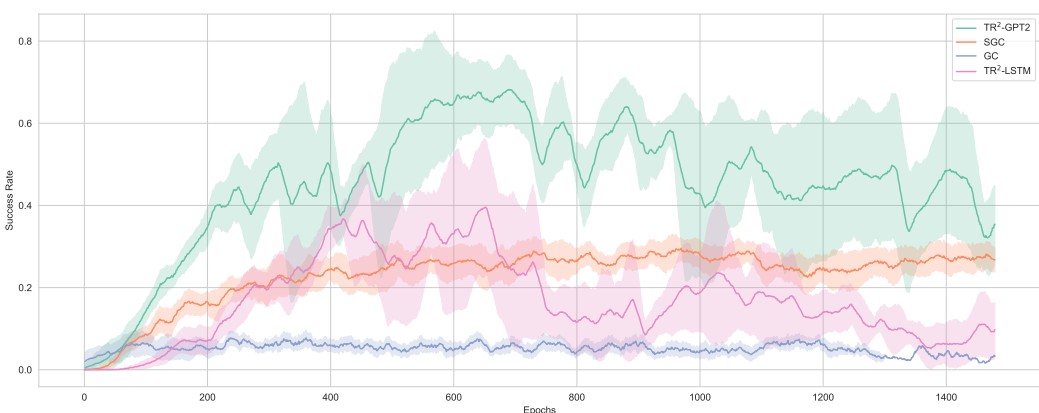

Figure 17: Couch Moving (Online) Training Success Rates

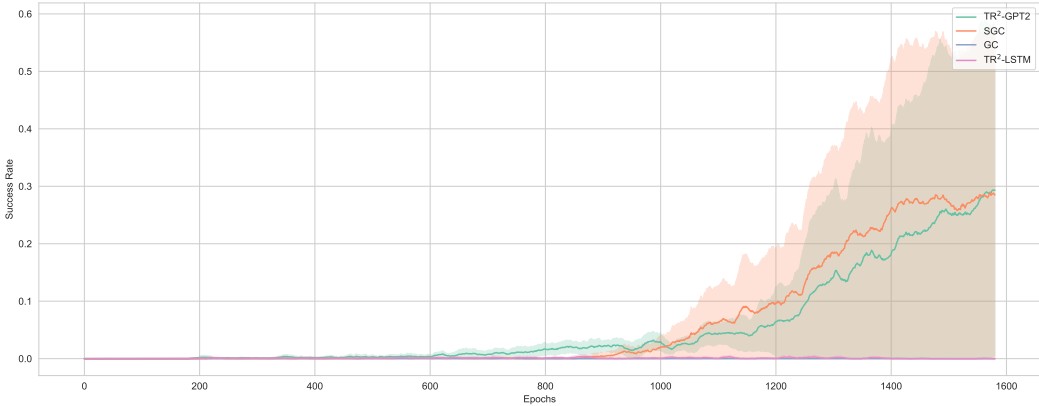

Figure 18: Block Stacking Training Success Rates

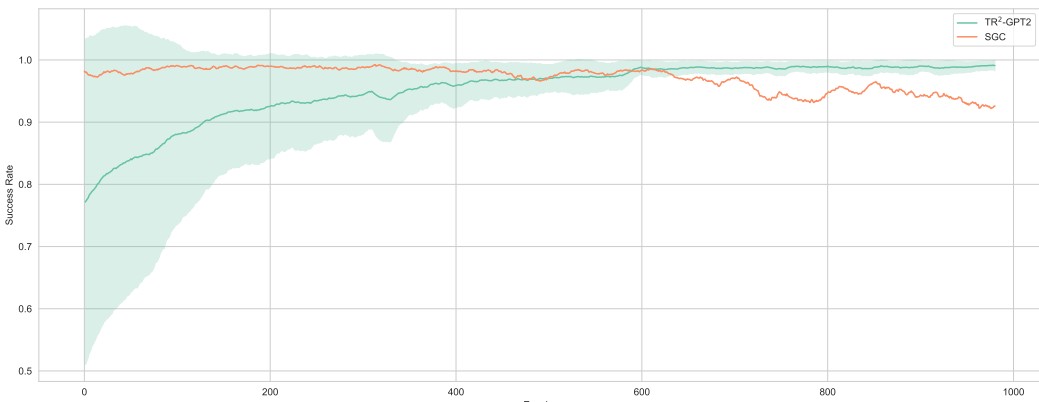

Figure 19: Block Stacking Training Success Rates with gradient accumulation turned on, continuing training from the training in Fig. 18. Note that SGC does not have an error bar since only one of the tested seeds was able to finetune and get a substantial success rate.

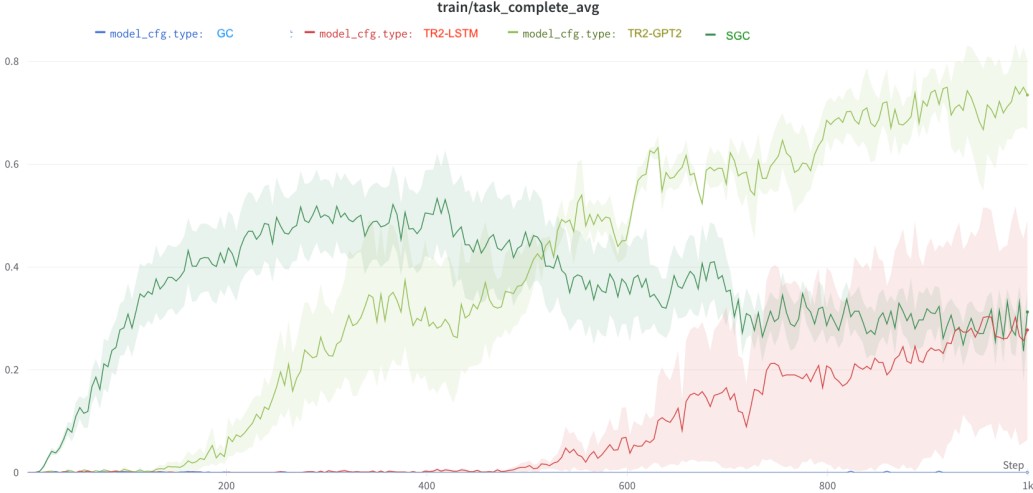

Figure 20: Open Drawer Success Rates. Note that the training runs for GC are displayed but since it gets 0 success rate almost all the time it is difficult to see the runs at the bottom.

