# OpenReview forum: "Abstract-to-Executable Trajectory Translation for One-Shot Task Generalization"
_ICLR.cc/2023/Conference — Submitted to ICLR 2023_

### Official Review · Reviewer_iXws · 2022-10-24

**Confidence:** 4
**Correctness:** 3
**Technical Novelty And Significance:** 3
**Empirical Novelty And Significance:** 3
**Recommendation:** 6

**Clarity, Quality, Novelty And Reproducibility:**

### Quality
This paper is well-written and easy to follow.

### Clarity
There are some unclear points in the paper:
- In figure 4 (a), the policy pays attention to the current and the next next chamber, but not to the next chamber where the agent needs to rotate the couch. In contrast, in (b), the policy pays attention to the next chamber, although the agent doesn't need to rotate the couch anymore. Also, I guess the reason why the policy doesn't pay attention to the past paths might be simply because of the context length of the executable policy. I'm not sure whether these patterns describe why TR$^2$-GPT2 works well.

### Originality
The framework that leverages abstract trajectory in the simplified simulator and translates it with a causal transformer to the executable actions seems a novel approach for task generalization to unseen configurations.


**Strength And Weaknesses:**

### Strength
- The proposed framework, trajectory translation, seems to be a novel paradigm for "demonstration-guided" reinforcement learning. Borrowing the success of causal Transformer architecture, the policy can "translate" an ideal/simplified trajectory to an actionable one in unseen configurations.
- The experimental results show TR$^2$-GPT2 significantly outperforms other baselines (TR$^2$-LSTM, SGC, GC) and could solve complex tasks, such as Block Stacking or Open Drawer.

### Weaknesses
- It seems unclear how we can obtain abstract trajectories.  If we collect abstract trajectories with some manually-designed waypoints, some robotics planners could solve the tasks even in executable settings.
- It is also unclear that how the proposed methods handle different dimensionality of the state between abstract and executable trajectory. The dimension of $s^H$ and $s^L$ seems very different (pointmass vs robot).

**Summary Of The Paper:**

This paper proposes a novel online RL framework, Trajectory Translation, that enables one-shot task generalization in robotics with abstracted trajectories. Trajectory Translation takes an abstract trajectory that is obtained on the simplified simulator (pointmass, magnetic manipulation, etc) as an input combined with the current execution trajectory, and then policy outputs the executable actions. Such translation is learned with PPO and trajectory-aware state-distance-based reward function. Combined with GPT-2 architecture, the proposed method provides strong one-shot performances on Box-Pusher/Couch Moving/Block Stacking/Open Drawer. In difficult or some intervention tasks, the replanning with the abstracted instruction improves the performances.

**Summary Of The Review:**

While there are some unclear points listed above, the proposed method seems a decent contribution to the community (novelty, good demonstration video, etc). In addition, the empirical evaluation is persuasive enough. Considering those aspects, I lean toward acceptance.

---
Update: I thank the authors for addressing my concerns and questions. Even after the discussion period, I still think my evaluation is fair for your contributions for now. So, I'd like to keep my score as it is.

---

> ### Author Response · Authors · 2022-11-08
> **Response to reviewer iXws**
>
> Thank you for taking the time to review our paper. We’re happy to hear that you found our work well-written and our supplementary materials persuasive. We address some of the weaknesses/concerns below.
>
> > It seems unclear how we can obtain abstract trajectories. If we collect abstract trajectories with manually-designed waypoints, some robotics planners could solve the tasks even in executable settings.
>
> Abstract trajectories are effectively a sequence of heuristically selected waypoints (but not actions) for where the agent should move and how the world should change (e.g., where blocks are displaced to). In our work, we use heuristics to generate abstract trajectories, which is permissible as the high-level space is simplified, and we simply just move 3D points around in space.
>
> Some robotics planners might be able to solve the tasks given the same waypoints. However, they need **explicit models of the environment and the agent** at hand to execute the abstract trajectory / follow waypoints, **requiring much more human engineering efforts than simply providing waypoints**. In contrast, our method leverages reinforcement learning to bridge the gap between abstract and executable trajectories automatically. As a result, our approach would allow one to solve the same tasks using different robot arms and geometries (demonstrated in Open Drawer test environments) without extra engineering, just switching out the models in the simulator.
>
> > It is also unclear that how the proposed methods handle different dimensionality of the state between abstract and executable trajectory. The dimension of $s^H$ and $s^L$ seems very different (pointmass vs robot).
>
> There are two notable components in which the difference in dimensionality can pose a problem: processing high-level and low-level states together in the transformer and the reward function design.
>
> Transformer:
> The high-level states in the abstract trajectory and low-level states in the executable trajectory do not necessarily need to be in the same space or share dimensions. This is achieved by **using a different MLP-based encoder to encode different modalities into the same latent space**, akin to Decision Transformer by Chen et al. [1]. Then once the two modalities (high vs. low-level states) are encoded into the same latent space, we can concatenate them as a whole sequence and pass them through the transformer to process.
>
> Reward Function Design:
> Our reward function is designed to encourage the low-level agent to follow the abstract trajectory as closely as possible. The notion of closeness is measured by a simple distance function $d(s^L, s^H) = ||f(s^L) - s^H||$. Importantly, the **mapping function $f$ bridges the two modalities** as it maps any low-level state $s^L$ to its equivalent high-level state $s^H$, **which is possible since we represent all objects in all our environments by their 3D position**. The rest of the reward function then forms a natural curriculum as the low-level agent is rewarded for matching a new high-level state in the abstract trajectory that is farther than before, ultimately encouraging the agent to match as many high-level states as possible.
>
>
>
> [1] Chen et al., Decision Transformer: Reinforcement Learning via Sequence Modeling. NeurIPS 2021
>
> > In figure 4 (a), the policy pays attention to the current and the next next chamber, but not to the next chamber where the agent needs to rotate the couch. In contrast, in (b), the policy pays attention to the next chamber, although the agent doesn't need to rotate the couch anymore. Also, I guess the reason why the policy doesn't pay attention to the past paths might be simply because of the context length of the executable policy. I'm not sure whether these patterns describe why TR-GPT2 works well.
>
> Figure 4 (a) and (b) are consistent in that the agent always looks forward up to two chambers ahead. Both the next chamber and next next chamber locations indicate the orientation of the upcoming corner, whereas attending to the location of anything further may be misleading.
>
> The context length is always long enough to include the entire abstract trajectory (which is effectively the map). At each step, the policy is given the entire abstract trajectory as part of its context, so it is permitted to attend to past paths. **Importantly, the policy learns to pay more attention to future locations on the map, indicating it learned that past parts are uninformative.** We will revise the paper to clarify this comment and further support our claims on the learned attention patterns.
>
> We hope the responses above clarify any misunderstandings. In light of our responses, would you be willing to increase your score, and if not, what do you think still needs revision or improvement?

---

> > ### Comment · Reviewer_iXws · 2022-11-13
> > **Response**
> >
> > I appreciate your responding to the review and providing detailed descriptions.
> >
> > **> comment on answer 1 (waypoint)**
> >
> > The results that the abstracted trajectory (by removing complex dynamics or physics) could work as a prompt for RL seem interesting. My concern here is I'm not sure how the cost is reduced between conventional robotics planners and the proposed approach. Even in both approaches or settings (sim or real), we always require the explicit model of the agents and environments (to spawn to the simulator, to command via ROS, etc). I agree that the proposed approach doesn't require it as an input to the module, but it requires massive online try-and-error instead. I think this is as costly as the planner approach. As RL and Transformer-application paper, I don't think this is not such a negative point. But I'd like to note that a similar approach (waypoint controller for robotics planner) has been established.
> >
> >
> > **> comment on answer 2 (high-level/low-level observation)**
> >
> > I thank the reviewer for clarifying the tokenization of observations with different dimensionality, and the mapping function for 3D coordinate-based reward function. One (possibly) relevant question is what is the architecture for the value function of PPO? Do you also use GPT-2 for the value function? It might better include such details.
> >
> >
> > **> comment on answer 3 (attention map)**
> >
> > The revised description seems to be clearer than before. Another question I came up with is how you obtained such an attention map from GPT-2? I imagined a square-size attention map (# of tokens x # of tokens), but it is not such a shape. Also, GPT-2 has a causal mask (only cares about the past) so it might not be intuitive that the proposed methods only focus on the "future" states.

---

> > > ### Author Response · Authors · 2022-11-16
> > > **Author's Response**
> > >
> > > Thanks for the quick follow-up! We appreciate the timeliness and address your comments below (we tried to limit to one post so we didn't copy paste the first two comments)
> > >
> > > > comment 1
> > >
> > > Thanks for pointing out the method of waypoint controllers for robotics planners. We agree that it is not exactly a negative point since we propose a separate learning-based approach.
> > >
> > > We wish to emphasize, though that since our approach works **under a weaker assumption (no env models as input)**, it is not completely fair to compare the online training cost of RL with robotics planners. The general approach of one/few-shot imitation approaches trained by RL, like our work, has been actively explored recently by multiple works, e.g. [1] [2] [3].
> > >
> > > Moreover we wish to emphasize that our appoarch is much faster than conventional robotic planners **during test time**.  In our method, we only need to train and do massive trial and error **once**. At test time, the policy can be deployed to various tasks starting from different initial states, and the trajecotries generated by the policy’s forward pass is cheap. However, robotics planners must always search/optimize a new trajecory for each different initial state or goal state at test time, even if given the waypoints ahead of time. If the waypoints are too sparse, robotics planners will perform poorly and run slowly, whereas with RL, while it may take longer to train, it will be fast at test time.
> > >
> > > [1] Zhu et al., Reinforcement and Imitation Learning for Diverse Visuomotor Skills. In RSS, 2018.
> > > [2] Lee et al., To follow or not to follow: Selective imitation learning from observations. In CoRL, 2019.
> > > [3] Zhou et al., Watch, Try, Learn: Meta-Learning from Demonstrations and Rewards. In ICLR, 2020
> > >
> > > > comment 2
> > >
> > > The actor and critic use the same architecture but do not share weights. We will update our submission to include this detail to help encourage reproducibility (and clarify in our officially released code later as well), thanks for pointing it out!
> > >
> > > > The revised description seems to be clearer than before. Another question I came up with is how you obtained such an attention map from GPT-2? I imagined a square-size attention map (# of tokens x # of tokens), but it is not such a shape.
> > >
> > > When predicting the final output token $z_{n+k}$, which is then fed into an MLP to generate an action (see Fig 2.), $z_{n+k}$ can attend to all previous tokens $x_1,...,x_n,x_{n+1},...,x_{n+k}$ which include the encodings of each high-level state in the abstract trajectory. The attention over each high-level state has some scalar value associated with it (mean attention of all heads) and the resulting triangle attention matrix is of shape $(n+k) \times (n+k)$. In Couch Moving, each high-level state is the high-level agent’s position on the map, which is just a point. So an abstract trajectory is composed of a sequence of these positions, which in turn is a map representation. This means at each step during inference, we have the attention value of each high-level state and equivalently, the attention value of each location on the map, allowing us to visualize the attention in a natural way. Implementation wise we do this by extracting the last row in the square attention matrix, which is a $1 \times (n+k)$ vector, and then removing the attention over past low-level states leaving behind a $1 \times n$ vector. We then map that vector over the couch moving map.
> > >
> > > Lastly, building upon your previous question about architecture, since we don’t write it in the submission, we clarify here that it is the actor’s attention that is being shown, not the critic's. We will update the submission to reflect this.
> > >
> > > > Also, GPT-2 has a causal mask (only cares about the past) so it might not be intuitive that the proposed methods only focus on the "future" states.
> > >
> > > From an empirical perspective, we don’t intend to claim that it only focuses on “future” states. The picture of the attention on the couch moving map shows that the agent does attend to some “past states”, just to a lesser degree.
> > >
> > > To clarify, by future, we mean future high-level states in the abstract trajectory, **which is in the “past” of the input sequence** as it forms the prompt. Future here refers to locations up ahead of the agent on the map closer to the goal; likewise, past refers to locations behind the agent in the map. From a writing perspective, we agree that using the term “future” in our original rebuttal response is confusing and unintuitive. We originally used the term in our response to emphasize the long-term dependency nature in our approach’s context to convey the reasoning behind transformers better, but we realize this can be erroneous and misleading.
> > >
> > > We hope these answer your questions, and if you have any more questions, we will be happy to answer them. Thank you!

---

> > > > ### Comment · Reviewer_iXws · 2022-11-17
> > > > **Response**
> > > >
> > > > I thank the authors for addressing my concerns and questions. I don't have further questions, and I think my evaluation is fair for your contributions for now. Let me keep discussing with other reviewers.
> > > >
> > > > For paper length, you may better follow 9-page limit, since CFP doesn't mention extra pages after the discussion period. I'm not a referee for that, but kindly recommend it to you.

---

### Official Review · Reviewer_QFp8 · 2022-10-24

**Confidence:** 3
**Correctness:** 3
**Technical Novelty And Significance:** 2
**Empirical Novelty And Significance:** 3
**Recommendation:** 6

**Clarity, Quality, Novelty And Reproducibility:**

The authors provide a flexible framework for integrating classical path planning algorithms into policies parameterized by neural networks. In doing so, the authors show that they provide high-level control by adjusting the "abstract" trajectories that the policy conditions on. I found the paper well-written and clear. I don't feel completely confident commenting on the originality of the work, but I do feel that this paper primarily uses standard tools (e.g. Seq2Seq models) but combines them in a way to get impressive task-generalization that I haven't seen from other papers.

**Strength And Weaknesses:**

I found this paper and the accompanying website and supplementary material easy and interesting to read. My personal pitch for the method is that goal-conditioning and reward-densification are both important for training a useful motion planner, and the authors use a high-level trajectory to enable both of these capabilities. The baselines also seem to demonstrate the importance of both of these features of the method, e.g. my interpretation is that SGC uses just reward-densification and GC/LSTM ablate the goal-conditioning mechanism to some extent. Additionally, I appreciate the section of the appendix that covers failure modes.

The main weakness that I see in this paper is I feel it would be good to get some intuition on what is and isn't a good abstract MDP to use for conditioning. Is there a way the authors can adjust the extent to which the abstract MDP is similar to the real MDP and show that the "closer" the abstract MDP is to the real MDP, the better the policy performs? What happens when the abstract MDP is equivalent to the actual MDP - is that oracle performance? Why do the authors condition on abstract states instead of abstract actions? If the model conditioned on actions, the case where the abstract MDP is equivalent to the real MDP would just be a copying task but since it's states, it's less obvious to me whether or not using the real MDP would produce the best results.

**Summary Of The Paper:**

The authors tackle the problem of motion planning for robotics. Given a motion planning problem, the authors first simplify the problem to an "abstract" MDP that classical planning algorithms can solve. The authors then use a policy parameterized by a transformer that conditions on the solution to the "abstract" problem while taking action in the "real" MDP. The authors demonstrate that by choosing different "abstract" trajectories for the policy to condition on at test time, the authors can direct the robot to perform impressive novel tasks not seen during training.

**Summary Of The Review:**

This paper proposes a principled strategy for leveraging classical path planning algorithms to improve neural controllers. I found the numerous empirical results quite compelling. With additional analysis of which kinds of "abstract MDPs" are best for conditioning on, I think this paper will be a good contribution to the machine learning community.

---

> ### Author Response · Authors · 2022-11-08
> **Response to reviewer QFp8**
>
> Thank you for taking the time to review our submission and giving feedback. We’re excited to know that you thought our empirical results were compelling and found the accompanying material interesting and easy to follow. We address your weaknesses below:
>
> > The main weakness that I see in this paper is I feel it would be good to get some intuition on what is and isn't a good abstract MDP to use for conditioning. Is there a way the authors can adjust the extent to which the abstract MDP is similar to the real MDP and show that the "closer" the abstract MDP is to the real MDP, the better the policy performs?
>
> We agree that it would be an interesting problem to investigate and get insights into what abstract MDPs are suitable for generating abstract trajectories to condition policies on in the real MDP. However, it is difficult to quantify the distance between the abstract MDP and the real MDP, but we note that the purpose of the abstract MDP is **only** to generate an abstract trajectory for conditioning real policies. Therefore, adjusting the abstract MDP is equivalent to adjusting the abstract trajectory in our case. To this end, we perform an ablation study on abstract trajectories to study a similar problem to reveal some insights.
>
> In section 4.4, we run an ablation study by training different TR2-GPT2 models on different granularities of the abstract trajectory (which varies abstract trajectory length). On one end, with only a few high-level states (short abstract trajectory length), we show in figure 5 (right) that the performance is worse compared to providing more high-level states. On the other hand, with a granular enough abstract trajectory, it is informative enough to be learned and reasoned by TR2-GPT2 to get a good enough success rate. Therefore, as you guessed, the “closer” the abstract MDP is to the real MDP (here, we mean more high-level states), the better the policy performs.
>
> Another example is the fine-grained control of the robot arm. With sufficiently granular abstract trajectories, one can engineer an abstract trajectory to specify more precise paths for the robot arm to follow. As a result, in Block stacking, we can easily control a few factors, such as how high the block is picked up and the angle of approach to the goal location, which is useful for skills such as collision avoidance. If it were not sufficiently granular (e.g. extremely abstract with very few high-level states), then this would not be possible.
>
> > What happens when the abstract MDP is equivalent to the actual MDP - is that oracle performance?
>
> If the abstract MDP is strictly equal to the real MDP, then it would be better to condition on abstract actions, and it's simply a copying task. However, the core contribution of our approach is to build an abstract MDP that removes the difficult features of the real MDP, like physical manipulation and contact simulation. If the abstract MDP is equivalent to the real MDP, it makes the abstract trajectory generation as hard as the original problem. Therefore, it does not make sense to use an abstract MDP equivalent to the real MDP in our case.
>
> >  Why do the authors condition on abstract states instead of abstract actions? If the model conditioned on actions, the case where the abstract MDP is equivalent to the real MDP would just be a copying task but since it's states, it's less obvious to me whether or not using the real MDP would produce the best results.
>
> Since the abstract MDP isn’t equivalent to the real MDP, it makes more sense to condition on abstract states than actions. Concretely, we use the abstract MDP to help describe the concept of abstract trajectories, but in code, there are no abstract actions since the abstract trajectory is generated by teleporting point masses around to solve a task. A sequence of abstract states provides waypoints a low-level policy should try and follow to solve the desired task.
>
> We hope we have addressed your concerns. Given the clarifications, would you be willing to increase your score? If there are still additional concerns or weaknesses, we are happy to discuss them and make appropriate revisions.

---

> > ### Comment · Reviewer_QFp8 · 2022-12-02
> > **Thank you for the response**
> >
> > Thank you for the authors' response. I do think this paper demonstrates a simple and effective way to leverage our knowledge of solutions to simplified planning problems in order to solve harder planning problems. Without being an expert in this field, I don't feel confident enough to increase my score, but I do believe I learned something from reading this paper about one-shot RL and feel this paper should be accepted.
> >
> > I'd additionally like to argue again that the authors ablate more the choice of specific "abstract problem" to use for conditioning. For someone implementing the ideas from this paper, the first question is what abstract problem should be chosen. The new experiments on abstract trajectory granularity are interesting, but they don't provide a sense of how to choose the actual abstract MDP.

---

> > > ### Author Response · Authors · 2022-12-04
> > > **Author's Response**
> > >
> > > Thanks for responding to our rebuttal, we appreciate the comments and clarity over what you find useful about our submission. We recognize that we might not have the ideal insightful experiments with regard to the choice of abstract MDP. We do want to reiterate (for other reviewers as well) that the simple point-mass-based abstract MDP works well for our environments, so we investigated other more quantifiable ablations.

---

> ### Author Response · Authors · 2022-11-23
> **Looking Forward to your Response!**
>
> Dear reviewer QFP8,
>
> Given the discussion phase is quickly passing, we want to know if our response resolves your questions If you have any further questions, we are more than happy to discuss them. Thanks again for your valuable questions and feedback!

---

### Official Review · Reviewer_Nwsy · 2022-10-25

**Confidence:** 5
**Correctness:** 2
**Technical Novelty And Significance:** 3
**Empirical Novelty And Significance:** Not applicable
**Recommendation:** 5

**Clarity, Quality, Novelty And Reproducibility:**

The paper is written clearly. The experiments seem through. Most implementation details can be found in the appendix.

Figure 20 seems to be rendered with only 10 runs, which is supposed to be 12 runs.


**Strength And Weaknesses:**

### Strengths

* The proposed method can solve very long horizon tasks and demonstrate impressive results, including stacking 26 blocks out of 28 blocks in the real world.
* The experiments are exhaustive and the proposed method outperforms baselines in many experimental setups.
* The proposed method generalizes to unseen tasks.

### Weaknesses

* The proposed method requires a significant amount of domain knowledge about the "environment" and "task", including an abstract environment, a mapping from low-level states to high-level states, a distance function between two high-level states, and a high-level solution trajectory in the abstract environment. The reviewer is wondering whether these are practical assumptions to make.
* Also, the proposed method requires almost similar efforts with providing sub-goals for the task directly in the original environments, which can make the problem much easier.
* The reward formulation considers the first similar state from the current state (or previous states) to measure the progress of the agent. This may not work when there are multiple similar states in the high-level trajectory, such as periodic tasks. This may require some temporal information in the state space, which makes the mapping to the abstracted environment infeasible. Discussion on this limitation would be appreciated.
* The training curves in Figure 16-20 show lower training success rates compared to the numbers presented in Table 1. This needs some explanation.
* Comparison to SILO on the main experiments would be appreciated with the properly tuned window size depending on the high-level plan's granularity.
* The real-world experiments show very impressive results in 28-block stacking. However, the simulated experiment shows much lower success rates in 5-block and 6-block stacking, which is likely a lot easier than the real-world 28-block stacking task. The authors can explain the discrepancy between the two results in the paper.

**Summary Of The Paper:**

This paper proposes to tackle a long-horizon task by (1) designing a simplified version of the environment, (2) solving the task in the simplified environment, and (3) translating the solution trajectory in the simplified environment into the trajectory in the original environment. By decomposing high-level planning and low-level task completion, the proposed method could solve four long-horizon tasks.

**Summary Of The Review:**

The paper proposes to replace high-level planning with a manually-designed simplified environment and a solution from the simplified environment. Although the paper shows the benefits of the proposed approach, given that it highly relies on the hand-designed high-level planner, it seems to have limited scalability. Therefore, I would be leaning toward a weak rejection.

---

> ### Author Response · Authors · 2022-11-09
> **Response to reviewer Nwsy (1/2)**
>
> Thank you for taking the time to review our paper. We’re happy that you found our paper well-written and see the benefits of our approach. We appreciate the comprehensive list of weaknesses, and we address them below:
>
> > The proposed method requires a significant amount of domain knowledge about the "environment" and "task," including an abstract environment, a mapping from low-level states to high-level states, a distance function between two high-level states, and a high-level solution trajectory in the abstract environment. The reviewer is wondering whether these are practical assumptions to make.
>
> These assumptions are practical in our setting and can be satisfied by our general construction of the high-level state space. Concretely, for **all environments**, we represent the high-level agent and all objects as **point masses**. By using this abstraction:
> - The mapping function $f$ from low-level states to high-level states simply removes every dimension but 3D position details.
> - The distance function is always $d(s^L, s^H) = ||f(s^L) - s^H||$ where $f$ is the mapping function from before.
> - The heuristic that generates the high-level solution trajectory simply teleports point masses around in space to represent a plan for a task.
>
> Thus, all assumptions are practical, and significant domain knowledge is not required for the abstract environment, the mapping function, or the distance function since they are always the same. The only domain knowledge required is the task itself and being able to draw out a simple plan of how objects should move to solve a task. This kind of domain knowledge is already applied to many other approaches in one-shot imitation learning, such as using domain knowledge to physically demonstrate a task in a video or engineering dense rewards. See this timestamp of our supplementary video for visual examples of the high-level trajectory, which demonstrate its simplicity: https://www.youtube.com/watch?v=02nnkiMeDeA&t=278s.
>
> Furthermore, we will revise the paper to clarify our choice of abstraction and address scalability.
>
> > Also, the proposed method requires almost similar efforts with providing sub-goals for the task directly in the original environments, which can make the problem much easier.
>
> Directly providing sub-goals and training on them is **insufficient** in our cases. We benchmark against providing sub-goals via the SGC (subgoal-conditioned policy) baseline in Table 1 and show the success rates are worse than TR^2-GPT2. One reason for the lower success rates of the subgoal-conditioned policy is due to long-horizon dependencies. This is exemplified by the Couch Moving environment, where a single sub-goal is insufficient in helping determine which orientation to rotate. While one could tune the selection of sub-goals for the SGC baseline, TR^2-GPT2 does this automatically as TR^2-GPT2 attends to the entire abstract trajectory (which can be seen as a sequence of sub-goals).
>
> > The reward formulation considers the first similar state from the current state (or previous states) to measure the progress of the agent. This may not work when there are multiple similar states in the high-level trajectory, such as periodic tasks. This may require some temporal information in the state space, which makes the mapping to the abstracted environment infeasible. Discussion on this limitation would be appreciated.
>
> Thanks for bringing this up. Indeed the current reward function formulation would not be able to handle situations with similar high-level states, such as periodic tasks. Instead of incorporating temporal information, one way to address this limitation would be to do a similar process we do with block stacking multiple blocks where we chunk the abstract trajectory into smaller parts. In this case, we would chunk it such that no high-level state is repeated. We’ve uploaded a video demonstrating a successful example where the task is repeatedly picking and placing a block between two locations three times: https://youtube.com/shorts/nn4QmrGf-GQ?feature=share. While the overall abstract trajectory has repeated high-level states, we can overcome the issue of similar high-level states by chunking it appropriately (e.g., whenever the robot arm returns to its rest position).
>
> We will add this discussion about this point in the updated manuscript, thanks for the suggestion!
>
> > The training curves in Figure 16-20 show lower training success rates compared to the numbers presented in Table 1. This needs some explanation.
>
> The training curves show the success rate of using the gaussian policy in PPO, which has exploration noise leading to lower success rates. During evaluation, we don’t sample from the gaussian and use the mean of the gaussian instead. Thus, evaluation results in table 1 are generally higher.

---

> > ### Author Response · Authors · 2022-11-09
> > **Response to reviewer Nwsy (2/2)**
> >
> > > Comparison to SILO on the main experiments would be appreciated with the properly tuned window size depending on the high-level plan's granularity.
> >
> > We planned on benchmarking SILO on our environments and tuning the window size hyperparameter, but the code of SILO is not released. As a compromise, we have benchmarked our approach on re-implementations of SILO’s environments to show that our approach outperforms SILO. See section 4.2 and table 2 for results.
> >
> > > The real-world experiments show very impressive results in 28-block stacking. However, the simulated experiment shows much lower success rates in 5-block and 6-block stacking, which is likely easier than the real-world 28-block stacking task. The authors can explain the discrepancy between the two results in the paper.
> >
> > The failure reason for stacking a tower of height 5 or 6 is stacking a *high* tower is very challenging since, in training, the model only stacks blocks up to height 3. The 28-block stacking configuration only has blocks up to height 4, and our model has shown a high success rate on stacking towers of height 4, so 28-blocks being stacked is not difficult and further demonstrates the strong long-horizon capabilities.
> >
> > > Figure 20 seems to be rendered with only 10 runs, which is supposed to be 12 runs.
> >
> > If we understand correctly, the 10 runs refer to 3 runs for each TR^2-GPT2, TR^2-LSTM, and SGC, and 1 run of GC. While it may appear there is only one run of GC, we actually display 3 runs of GC but it has 0 success rate almost all the time, so the line is at the bottom and the standard deviation barely shows.
> >
> > In light of our clarifications, would you consider increasing the score for our submission? If not, let us know what additional changes you think are necessary for this work to be accepted. Thanks so much for your time!

---

> > > ### Comment · Reviewer_Nwsy · 2022-11-17
> > > **Response to authors**
> > >
> > > Thank you for your response. Many of my questions and suggestions are addressed in the rebuttal.
> > >
> > > * The abstraction with point masses works well in the experiments as most abstract trajectories are designed to be useful. However, one can easily come up with tasks and environments where this assumption does not hold. For example, if a maze for the "Couch Moving" task has multiple possible routes to the goal position and not all the routes are feasible due to the shape of the couch and the maze, like too narrow passage. Then, the success of the proposed method will be highly dependent on whether a feasible path is chosen as an abstract trajectory or not. If an infeasible path is chosen as an abstract trajectory, the method is likely to fail. Thus, to get a good abstract trajectory, we need further assumptions or domain knowledge about environments and tasks. Unless this point-mass abstraction can be applied to a significant amount of applications, this seems like a critical limitation of the proposed method.
> > >
> > > * My claim was about providing nearly exact sub-goals manually or with significant domain knowledge, rather than predicting sub-goals as in the SGC baseline. Designing abstract trajectories seems to require as much effort as providing waypoints in the original environments, or the difference is not significant.
> > >
> > > * As explained by the authors, it may not affect the plot but I noticed "Showing first 10 runs" under the plot title of Figure 20.
> > >
> > > As the first two are my major concerns about the paper, I would stay with my original judgment, weak rejection.

---

> > > > ### Author Response · Authors · 2022-11-18
> > > > **Response to Reviewer Nwsy (1/2)**
> > > >
> > > > Thanks for your comprehensive response, we address them further below. We want to clarify some misunderstandings and emphasize how our core contributions bring benefits that improve over previous approaches that also require similar amounts of task-specific domain knowledge.
> > > >
> > > > > The abstraction with point masses works well in the experiments as most abstract trajectories are designed to be useful. However, one can easily come up with tasks and environments where this assumption does not hold. For example, if a maze for the "Couch Moving" task has multiple possible routes to the goal position and not all the routes are feasible due to the shape of the couch and the maze, like too narrow passage. Then, the success of the proposed method will be highly dependent on whether a feasible path is chosen as an abstract trajectory or not. If an infeasible path is chosen as an abstract trajectory, the method is likely to fail. Thus, to get a good abstract trajectory, we need further assumptions or domain knowledge about environments and tasks. Unless this point-mass abstraction can be applied to a significant amount of applications, this seems like a critical limitation of the proposed method.
> > > >
> > > > Most one-shot imitation learning methods require some sort of task-specific domain knowledge for demonstrations at test time, whether it's a human video [1] [2], or a low-level demo [3], **so this is a limitation for almost all methods in this research direction**. By designing a more flexible and efficient way to provide guidance and a strong translator to utilize the guidance, our approach relying on the same domain knowledge can scale better on long-horizon and complex tasks.
> > > >
> > > > We also like to point out that the proposed example asks the agent to complete the task with a demonstration that contains significant infeasible portions for the agent, **which is not our exact scenario and would be a limitation of almost all one-shot IL methods.** Despite this, we do show that **our model can still handle this kind of issue** when we show that our approach improves on SILO [2] on the environments from SILO, which tasks agents do not strictly follow the demonstration and skip infeasible portions (such as when the demonstration goes through obstacles in SILO’s Obstacle Push and Pick-and-place environments).
> > > >
> > > > Finally, we reiterate how our approach improves upon previous one-shot IL work while relying on the same task-specific domain knowledge.
> > > >
> > > > **Scaling to long-horizon tasks:** Human and low-level demos become less feasible to collect for longer tasks. For human demos, it’s difficult since more time is required to demonstrate a long-horizon task, such as stacking the 28 blocks in the castle configuration.
> > > >
> > > > **Scaling to complex tasks:** For more difficult tasks, sometimes it is infeasible to have a human execute that task due to its difficulty. One example could be opening a kitchen cabinet high above the ground, which can’t be demonstrated easily by people who can’t reach it. However, one can easily program a heuristic to generate an abstract trajectory to solve it easily and leverage the robot's abilities.
> > > >
> > > > **Re-planning:** Since we use abstract trajectories that are easily generated by a heuristic (requiring similar domain knowledge to previous approaches), we can re-generate abstract trajectories at test time to instantly change the behavior and direct the low-level robot to do a wide variety of tasks. Comparatively, with human or low-level demos, the engineering effort to handle these failure cases or deal with interventions lies on the human, which is easy to do, but not scalable as we cannot always assume a human is there to demonstrate and fix mistakes. Instead with abstract trajectories, the engineering effort lies on the programmer who can engineer plans to handle failure cases that can then be repeatable without human supervision in the future.
> > > >
> > > >
> > > > [1] Yu et al., One-Shot Imitation from Observing Humans via Domain-Adaptive Meta-Learning. In RSS 2018.
> > > >
> > > > [2] Y. Lee et al., To follow or not to follow: Selective imitation learning from observations. In CoRL, 2019.
> > > >
> > > > [3] Mandi et al., Towards More Generalizable One-shot Visual Imitation Learning. In ICRA 2022.

---

> > > > > ### Author Response · Authors · 2022-11-18
> > > > > **Response to Reviewer Nwsy (2/2)**
> > > > >
> > > > > > My claim was about providing nearly exact sub-goals manually or with significant domain knowledge, rather than predicting sub-goals as in the SGC baseline. Designing abstract trajectories seems to require as much effort as providing waypoints in the original environments, or the difference is not significant.
> > > > >
> > > > > We would like to clarify that the SGC baseline **does not predict sub-goals**. Instead, **it is  “providing nearly exact sub-goals manually,” as you said**. The sub-goals are selected from the abstract trajectory’s high-level states by picking every $n$th high-level state and providing the next high-level state as a sub-goal. We have tuned this $n$ per task to make this SGC baseline as strong as possible but note that this tuning is not trivial. Therefore, the results of SGC indicate that trivially utilizing sub-goals does not work very well.
> > > > >
> > > > > Indeed writing a heuristic to construct waypoints is similar to writing a heuristic to generate abstract trajectories. However, we would like to emphasize that a core contribution of our approach is that we outperform other baselines (e.g., providing sub-goals manually and directly feeding them to the policy), which requires the same human effort/domain knowledge. Our approach outperforms SGC on test tasks thanks to the transformer architecture, which attends and processes the entire abstract trajectory and generalizes better. We further outperform SILO [2] in some of their environments, showing the benefit of our approach.
> > > > >
> > > > > > As explained by the authors, it may not affect the plot but I noticed "Showing first 10 runs" under the plot title of Figure 20.
> > > > >
> > > > > Ah, we see where the misunderstanding comes from. To clarify, there are 12 runs grouped into 4 groups, one for each model. “Showing first 10 runs” means showing the first 10 groups (which includes all 4 groups). We will remove this line from the figure to remove any confusion.
> > > > >
> > > > > Thanks again for your time discussing with us, we hope that addresses your concerns and highlights why our approach to one-shot imitation learning has strong benefits compared to previous approaches that also require similar task-domain knowledge.
> > > > >
> > > > > [1] Yu et al., One-Shot Imitation from Observing Humans via Domain-Adaptive Meta-Learning. In RSS 2018.
> > > > >
> > > > > [2] Y. Lee et al., To follow or not to follow: Selective imitation learning from observations. In CoRL, 2019.
> > > > >
> > > > > [3] Mandi et al., Towards More Generalizable One-shot Visual Imitation Learning. In ICRA 2022.

---

### Official Review · Reviewer_KkrM · 2022-10-31

**Confidence:** 4
**Correctness:** 2
**Technical Novelty And Significance:** 2
**Empirical Novelty And Significance:** 3
**Recommendation:** 5

**Clarity, Quality, Novelty And Reproducibility:**

This paper is well motivated by solving one-shot task generalization. It propose to achieve the goal by decoupling plan generation and plan execution, such disentanglement is demonstrated useful in deployment. Borrowing the idea of seq2seq model in machine translation, TR2 solve the original tasks with abstract-to-executable trajectory translation. TR2 is also extensively evaluated on various settings and tasks.

However, there are a couple of technical things need to be further clarified. 1) unclear of the rational of abstract-to-execution policy translation. The key technical novelty of this work is to formulate the one-shot task generalization problem as a abstract-to-execution policy translation problem. However, I am not clear about the definition of 'abstract (high-level) trajectory' and 'execution (low-level) policy'. Based on box pusher example, seems the abstract trajectory is some trajectories with none-defined (no explicit) actions or dynamics, while these terminologies are not well defined. Seeing from the pipeline of TR2, the utilization of the abstract trajectory is more like producing a context-level prompt to perform execution. However, when using prompt to perform generalization, the assumption is that there are underlying correlation between the prompt and the target generation sequences. However, based on the illustration of the abstract and execution trajectories, seems they are lie in different dynamics distribution. In this case, I am confused how trajectories under different policies can help for generalization. Unless it is only used for generalize perception space (assuming they are in the same environment).
2) maybe a incorrectness of causal Transformer utilization, and lack of implementation details in transformer (e.g. tokenization, positional embeddings, attention masking, etc.). As I cannot find implementation details of the transformer, I can only figure out from Figure2 and Sec. 3.3. From these, seems the abstract trajectories and execution trajectories are first encoded into a sequence of token embeddings by separate encoders (i.e. x1, x2, ... xn; xn+1, xn+2... xn+k), and then they are fed into a GPT-2 Transformer to perform auto-regressive prediction. However, one thing strange here: as GPT2 is a causal transformer with masked attention, so the input sequence (x1, x2. ... xn, xn+1, xn+2, ... xn+k) is computed with sequence dependencies, i.e. x2 attn (x1, x2), xn attn(x1, x2, .. xn-1), ... xn+k attn(x1, x2, ... xn+k-1).  On the contrary, the encoder takes sequences parallelly. Then it is wired to me, e.g. you are predicting x2 by seeing x1 and x2, however, the x1 and x2 token embedding already seeing the future. The utilization of the Transformer needs to be further clarified.
3) lack of comparison with the other state-of-the-arts approaches.

**Strength And Weaknesses:**

Pros:
This paper is well motivated by solving one-shot task generalization. It propose to achieve the goal by decoupling plan generation and plan execution, such disentanglement is demonstrated useful in deployment. Borrowing the idea of seq2seq model in machine translation, TR2 solve the original tasks with abstract-to-executable trajectory translation. TR2 is also extensively evaluated on various settings and tasks.

Cons:
1) unclear of the rational of abstract-to-execution policy translation. The key technical novelty of this work is to formulate the one-shot task generalization problem as a abstract-to-execution policy translation problem. However, I am not clear about the definition of 'abstract (high-level) trajectory' and 'execution (low-level) policy'. Based on box pusher example, seems the abstract trajectory is some trajectories with none-defined (no explicit) actions or dynamics, while these terminologies are not well defined. Seeing from the pipeline of TR2, the utilization of the abstract trajectory is more like producing a context-level prompt to perform execution. However, when using prompt to perform generalization, the assumption is that there are underlying correlation between the prompt and the target generation sequences. However, based on the illustration of the abstract and execution trajectories, seems they are lie in different dynamics distribution. In this case, I am confused how trajectories under different policies can help for generalization. Unless it is only used for generalize perception space (assuming they are in the same environment).
2) maybe a incorrectness of causal Transformer utilization, and lack of implementation details in transformer (e.g. tokenization, positional embeddings, attention masking, etc.). As I cannot find implementation details of the transformer, I can only figure out from Figure2 and Sec. 3.3. From these, seems the abstract trajectories and execution trajectories are first encoded into a sequence of token embeddings by separate encoders (i.e. x1, x2, ... xn; xn+1, xn+2... xn+k), and then they are fed into a GPT-2 Transformer to perform auto-regressive prediction. However, one thing strange here: as GPT2 is a causal transformer with masked attention, so the input sequence (x1, x2. ... xn, xn+1, xn+2, ... xn+k) is computed with sequence dependencies, i.e. x2 attn (x1, x2), xn attn(x1, x2, .. xn-1), ... xn+k attn(x1, x2, ... xn+k-1).  On the contrary, the encoder takes sequences parallelly. Then it is wired to me, e.g. you are predicting x2 by seeing x1 and x2, however, the x1 and x2 token embedding already seeing the future. The utilization of the Transformer needs to be further clarified.
3) lack of comparison with the other state-of-the-arts approaches.

**Summary Of The Paper:**

This paper introduced Trajectory Translation (TR2) framework that seeks to train low-level policies by translating an abstract trajectory into executable actions. TR2 is designed to solve tasks in  three steps:  1) build a paired abstract environment by simplifying geometry and physics, generating abstract trajectories, and solve the original task by an abstract-to-executable trajectory translator. TR2 borrowed the idea of seq2seq model in machine translation to overcome the none one-one mapping issue between abstract and executable trajectories, so that to enable low-level policy to follow the abstract trajectory. Experiments are conducted on various unseen tasks with different robot embodiments.

**Summary Of The Review:**

This paper is well motivated by solving one-shot task generalization. It propose to achieve the goal by decoupling plan generation and plan execution, such disentanglement is demonstrated useful in deployment. Borrowing the idea of seq2seq model in machine translation, TR2 solve the original tasks with abstract-to-executable trajectory translation. TR2 is also extensively evaluated on various settings and tasks.
However, there are a couple of technical things need to be further clarified. 1) unclear of the rational of abstract-to-execution policy translation. The key technical novelty of this work is to formulate the one-shot task generalization problem as a abstract-to-execution policy translation problem. However, I am not clear about the definition of 'abstract (high-level) trajectory' and 'execution (low-level) policy'. Based on box pusher example, seems the abstract trajectory is some trajectories with none-defined (no explicit) actions or dynamics, while these terminologies are not well defined. Seeing from the pipeline of TR2, the utilization of the abstract trajectory is more like producing a context-level prompt to perform execution. However, when using prompt to perform generalization, the assumption is that there are underlying correlation between the prompt and the target generation sequences. However, based on the illustration of the abstract and execution trajectories, seems they are lie in different dynamics distribution. In this case, I am confused how trajectories under different policies can help for generalization. Unless it is only used for generalize perception space (assuming they are in the same environment).
2) maybe a incorrectness of causal Transformer utilization, and lack of implementation details in transformer (e.g. tokenization, positional embeddings, attention masking, etc.). As I cannot find implementation details of the transformer, I can only figure out from Figure2 and Sec. 3.3. From these, seems the abstract trajectories and execution trajectories are first encoded into a sequence of token embeddings by separate encoders (i.e. x1, x2, ... xn; xn+1, xn+2... xn+k), and then they are fed into a GPT-2 Transformer to perform auto-regressive prediction. However, one thing strange here: as GPT2 is a causal transformer with masked attention, so the input sequence (x1, x2. ... xn, xn+1, xn+2, ... xn+k) is computed with sequence dependencies, i.e. x2 attn (x1, x2), xn attn(x1, x2, .. xn-1), ... xn+k attn(x1, x2, ... xn+k-1).  On the contrary, the encoder takes sequences parallelly. Then it is wired to me, e.g. you are predicting x2 by seeing x1 and x2, however, the x1 and x2 token embedding already seeing the future. The utilization of the Transformer needs to be further clarified.
This paper presents extensive evaluations and demo of deployments, I appreciated. While, it lacks of comparison with the other state-of-the-arts approaches. It is not convincing enough with such comparisons.

---

> ### Author Response · Authors · 2022-11-09
> **Response to reviewer KkrM (1/2)**
>
> Thank you for the constructive feedback and review. We will clarify some of the misunderstandings in our updated paper, and we address specific concerns below.
>
> > unclear of the rational of abstract-to-execution policy translation. The key technical novelty of this work is to formulate the one-shot task generalization problem as a abstract-to-execution policy translation problem. However, I am not clear about the definition of 'abstract (high-level) trajectory' and 'execution (low-level) policy'. Based on box pusher example, seems the abstract trajectory is some trajectories with none-defined (no explicit) actions or dynamics, while these terminologies are not well defined.
>
> To clarify, abstract trajectories are sequences of high-level states, of which **each high-level state is an abstract representation of relevant objects in an environment**. In all environments, the abstract representation is a **point mass**. For example, in the Block stacking task, the end-effector is represented as the end-effector's 3D position. The blocks are represented by their 3D positions. Thus, the high-level state is a 6D vector containing these two 3D positions. The low-level state information is from the actual environment (e.g., robot proprioception). As abstract trajectories are high-level, they are much easier to generate. To help visualize the abstract trajectories, you can see some visualizations of them in our supplementary video at this linked timestamp: https://youtu.be/02nnkiMeDeA?t=278.
>
> We also clarify that abstract trajectories and executable trajectories are different in that
> - They have different state representations (e.g. point mass vs robot proprioception)
> - Abstract trajectories do not explicitly contain actions (e.g., moving, close gripper), and the executable trajectory is the resulting trajectory between the agent’s interaction with the actual environment. While abstract trajectories do not contain actions, the positions of objects over time can implicitly describe actions: such as when both the end-effector’s 3D position and block’s 3D position move together, which describes grasping.
> - The abstract trajectory can vary in how sparse/spaced out subsequent high-level states are as dictated by the high-level agent. Executable trajectories are limited to the constraints of the actual action space.
>
> > Seeing from the pipeline of TR2, the utilization of the abstract trajectory is more like producing a context-level prompt to perform execution. However, when using prompt to perform generalization, the assumption is that there are underlying correlation between the prompt and the target generation sequences. However, based on the illustration of the abstract and execution trajectories, seems they are lie in different dynamics distribution. In this case, I am confused how trajectories under different policies can help for generalization. Unless it is only used for generalize perception space (assuming they are in the same environment).
>
> Indeed, the abstract trajectory can be viewed as a context-level prompt dictating the task. Moreover, there is a correlation between the prompt and the executed trajectory. Namely, **the prompt/abstract trajectory describes a general path to follow**, and the executed trajectory is produced to follow it as closely as possible via low-level actions. As a result, TR2-GPT2 learns to **follow the abstract trajectory as the task**. This way, at test time for unseen tasks, the agent only needs to follow the given abstract trajectory to complete the task without making high-level plans. That is why the abstract trajectory helps generalization.

---

> > ### Author Response · Authors · 2022-11-09
> > **Response to reviewer KkrM (2/2)**
> >
> > > maybe a incorrectness of causal Transformer utilization, and lack of implementation details in transformer (e.g. tokenization, positional embeddings, attention masking, etc.). As I cannot find implementation details of the transformer, I can only figure out from Figure2 and Sec. 3.3. From these, seems the abstract trajectories and execution trajectories are first encoded into a sequence of token embeddings by separate encoders (i.e. x1, x2, ... xn; xn+1, xn+2... xn+k), and then they are fed into a GPT-2 Transformer to perform auto-regressive prediction. However, one thing strange here: as GPT2 is a causal transformer with masked attention, so the input sequence (x1, x2. ... xn, xn+1, xn+2, ... xn+k) is computed with sequence dependencies, i.e. x2 attn (x1, x2), xn attn(x1, x2, .. xn-1), ... xn+k attn(x1, x2, ... xn+k-1). On the contrary, the encoder takes sequences parallelly. Then it is wired to me, e.g. you are predicting x2 by seeing x1 and x2, however, the x1 and x2 token embedding already seeing the future. The utilization of the Transformer needs to be further clarified.
> >
> > We do include additional implementation details in section E of the appendix. Still, we are glad to add more clarification and will update our submission to clarify some points. For code reference of the transformer usage, see our [anonymized link](https://github.com/abstract-to-executable/code/blob/main/skilltranslation/models/translation/translation_transformer.py).
> >
> > To clarify here, the transformer implementation code we used is adapted from Decision Transformers (DT) by Chen et al. [1]. As a result, we are **not** predicting the encoded tokens $x_{n+1}, …, x_{n+k}$, these are embeddings of the last $k$ observations given by the environment at timestep $t$. We are also **not** predicting $x_{1},...,x_{n}$, these embeddings are from the abstract trajectory itself, which forms the prompt/context and is always a part of the input sequence. So at each timestep $t$, the model is given the abstract trajectory $s^H_1, … s^H_{n}$ and the last $k$ observations $s^L_{t - k + 1}, …, s^L_{t}$, which are encoded as $x_1, … ,x_n$ and $x_{n+1},..., x_{n+k}$ respectively. The input to our transformer is then $x_{1},...,x_{n}, x_{n+1}, …, x_{n+k}$. The output of our transformer is the output embedding $z_{n+k}$, which is fed through an MLP to produce an action $a_t$ which is passed to the environment, and the environment will give us the next state $s^L_{t+1}$ and we repeat. Therefore, we always predict the action based on the abstract trajectory and past observations but never use future observations.
> >
> > > lack of comparison with the other state-of-the-arts approaches.
> >
> > As our approach presents the use of abstract trajectories, which to our knowledge, has not been done before (the closest is SILO [2] which we compare against in table 2), there are few approaches to benchmark against. We are interested in hearing about what other state-of-the-art approaches are publicly available that we should compare against. We would be happy to benchmark against them, post results here, and revise the paper to reflect the new benchmarking.
> >
> > Given the above clarifications and responses, would you consider raising the score for our paper? If not, could you let us know what revisions are necessary that you would like to see in order for this submission to be accepted? Thanks so much for your time!
> >
> > [1] Chen et al., Decision Transformer: Reinforcement Learning via Sequence Modeling. NeurIPS 2021
> >
> > [2] Y. Lee et al., To follow or not to follow: Selective imitation learning from observations. In CoRL, 2019.

---

> > > ### Comment · Reviewer_KkrM · 2022-12-01
> > > **Response to authors**
> > >
> > > Thanks for the authors response and clarification. Part of my questions and concerns are addressed in the rebuttal.
> > > While my main concern on the technical side is still remain. Since the key contribution of this work is to utilize the abstract trajectories and executable trajectories for decision making. However, such a manner lies in somewhere in between prompting with Transformer encoder-decoder pipeline. Without clear differentiation and empirical comparisons, the technical novelty is limited. On the other hand, I do agree reviewer Nwsy's point that the utilization of the abstract trajectories need further assumptions or domain knowledge about environments and tasks, which brings critical limitation of the proposed method.
> > >
> > > Considering these, I would raise my score to weak reject, which should be fair to your contribution.

---

> > > > ### Author Response · Authors · 2022-12-05
> > > > **Author's Response**
> > > >
> > > > Thanks for the response and for clarifying your current concerns.
> > > >
> > > > For clarity to all reviewers, we've provided a general response to the concern over the domain knowledge assumption in the main thread, which generally points out that this assumption is common with previous one-shot IL methods. Moreover, our approach, which shares the same assumptions on domain knowledge as previous one-shot IL methods, brings several benefits that previous methods cannot achieve.
> > > >
> > > > > While my main concern on the technical side is still remain.
> > > >
> > > > We are unsure which concerns on the technical side you still have. We are happy to address any specific concerns you have.
> > > >
> > > > > Since the key contribution of this work is to utilize the abstract trajectories and executable trajectories for decision making. However, such a manner lies in somewhere in between prompting with Transformer encoder-decoder pipeline. Without clear differentiation and empirical comparisons, the technical novelty is limited.
> > > >
> > > > Could you clarify what this means? We are confused by what “between prompting with Transformer encoder-decoder pipeline” means. Moreover, what differentiation and empirical comparisons do you refer to? We are happy to make them more clear and clarify details, e.g., our empirical comparisons with SILO [2].
> > > >
> > > > Otherwise, we want to emphasize that our contributions are novel and bring strong one-shot results on harder long-horizon tasks, and in the general comment, we clarify our three main contributions.

---

> ### Author Response · Authors · 2022-11-23
> **Looking forward to your response!**
>
> Dear reviewer Kkrm,
>
> Given the discussion phase is quickly passing, we want to know if our response resolves your questions If you have any further questions, we are more than happy to discuss them. Thanks again for your valuable questions and feedback!

---

### Author Response · Authors · 2022-11-09
**Summary of New Rebuttal Revision (updated 18 Nov 2022)**

We would like to thank all the reviewers for their in-depth, constructive feedback and for helping us improve the paper. We have submitted a revised PDF for the rebuttal, with new text added in red. For ease of access, the following details all changes and references which reviewer discussion the changes were based from

- Section 3.2: We clarify that we use the same process to make all abstract environments and emphasize that this is scalable since it's all based on simply mapping objects to point masses. See discussion with reviewer Nwsy
- Section 3.4: We add a short discussion on a limitation of the reward function when there are similar high-level states based on a discussion with reviewer Nwsy. We discuss more in-depth and show a successful example of overcoming the issue of similar high-level states in tasks such as periodic ones in section C.2.
- Section 4.5: We add a sentence clarifying that the agent receives the full abstract trajectory as input and is allowed to attend to any part of it, but learns to only attend to parts that are relevant and informative in Couch Moving. See discussion with reviewer iXws
- Section A.3: We add a comment on how success rates are low for stacking 5 or 6-block towers, but we can stack a 28-block castle configuration still. See discussion with reviewer Nwsy
- Section C.1: We add a line clarifying that the actor and critic share architecture but different weights. See discussion with reviewer ixWs
- Section E: We clarify further the auto-regressive nature of TR2-GPT2 and how it is adapted from Decision Transformers (DT) Chen et al., along with details on exactly the source of the input sequence to the transformer and how it relates to the environment. We also reference this section in section 3.2 for readers to easily jump to section E and read in-depth details. See discussion with reviewer KkrM.
- Section H: We add comments clarifying why the training curve’s success rates are sometimes lower than what is shown in the main results in Table 1, in addition to clarifying that Figure 20 indeed shows all training runs, but the GC baseline has too low of a success rate to be visible. Finally we remove a misleading text that says "showing 10 runs", the graph indeed shows all 12 runs (with 4 groups). See discussion with reviewer Nwsy.
- General: In order to stay within 9 page limit, we removed a few words and made a few things more concise. Moreover figure 1 has been  made smaller in order to get more space. (Thank you reviewer ixWs for pointing it out that there may be this limit still after phase 1!)

---

### Author Response · Authors · 2022-12-05
**General Rebuttal Response - Clarifications on contributions and summarizing rebuttal to main concerns**


Thank you to all the reviewers for the responses! We appreciate the constructive discussion and comments and want to keep things organized so for everyone to see we clarify our key contributions here as well as our rebuttal to the main concerns.

### **Key Contributions**

1. Introducing abstract trajectories for one-shot imitation learning. Abstract trajectories are easier to generate than other kinds of demonstrations (human video, low-level demos) thanks to the simplified abstract space (everything as a pointmass). Further, it can be feasibly generated at test time, unlike human videos or low-level demos, thus enabling re-planning.
2. Utilizing transformers to translate abstract trajectories into executable ones and bridging the high to low-level domain gap
3. Empirical results with solving unseen out-of-distribution tasks of horizons much longer than in training with high success rate.

### **Assumption of domain knowledge for environments/tasks:**
Reviewers KkrM and Nwsy both express concerns over the scalability of our proposed approach, as our policy is conditioned on a desired abstract trajectory to follow. While indeed abstract trajectories are simple and fast to generate, they require domain knowledge of how to solve a task to generate in the first place.

We would like to emphasize that **most one-shot imitation learning methods require some sort of task-specific domain knowledge for expert demonstrations at test time**, whether it's a human video [1] [2] (due to humans needing to know how to solve a task), or a low-level demo [3] [4] (due to dense-reward engineering or oracle policies), **so this is a limitation for almost all methods in this research direction, including recent publications.**

Furthermore, **while the same domain knowledge as previous one-shot IL methods is used, our approach brings features previous methods cannot achieve.** As pointed out by reviewer QFp8, our approach can leverage solutions to the simplified planning problem in the abstract environment (which can be solved via heuristics, RL with sparse rewards, simple motion planners) and is more scalable than previous methods that use solutions of other low-level robots or humans. Furthermore, since such abstract trajectories are easy to generate in contrast to videos or low-level demos, they can be regenerated on the fly to perform re-planning, an option previous one-shot IL methods do not have or would struggle to achieve. Finally, the results in our experiments demonstrate success on much longer-horizon tasks that further require generalization.

In summary, with a similar level of domain knowledge, our method enjoys better flexibility and achieves strong long-horizon task-generalization results while also providing previously infeasible features. We believe that improving the scalability is an important future direction of our work and the whole **one-shot IL** research field.


[1] Yu et al., One-Shot Imitation from Observing Humans via Domain-Adaptive Meta-Learning. In RSS 2018.

[2] Y. Lee et al., To follow or not to follow: Selective imitation learning from observations. In CoRL, 2019.

[3] Mandi et al., Towards More Generalizable One-shot Visual Imitation Learning. In ICRA 2022.

[4] Xu et al., Prompting Decision Transformer for Few-Shot Policy Generalization. In ICML 2022.

---

### Decision · Program_Chairs · 2023-01-20

**Decision:**

Reject

**Justification For Why Not Higher Score:**

Each reviewer acknowledged the authors' response. They point out that while the response and updates to the paper address some of the issues raised in the initial reviews, their primary concerns with the paper still hold.

**Justification For Why Not Lower Score:**

N/A

**Metareview: Summary, Strengths And Weaknesses:**

The paper considers the problem of generalizable long-horizon planning for embodied tasks such as robot manipulation. The paper proposes an approach that decouples the identification of a suitable long-horizon plan from its execution in the given environment. More specifically, the framework first establishes an abstract representation of the environment (e.g., where objects become point masses) where dynamics are ignored; solves for a high-level solution (trajectory) in this simplified representation of the environment; and then uses a sequence-to-sequence model to "translate" the high-level trajectories to their corresponding low-level trajectories in the actual environment. Experiments on both simulated and real-world robot manipulation tasks demonstrate that the proposed framework outperforms contemporary baselines.

The ability to perform long-horizon embodied (manipulation) tasks remains an open problem that is of significant interest to the robot learning community. As the reviewers point out, the idea of decomposing long-horizon tasks in terms of separate problems of plan generation and plan execution is sensible, as is the use of some amount of domain knowledge as a means of abstracting the environment and task. The specific way in which the proposed Trajectory Translation (TR2) framework achieves this decomposition, via reward densification and goal-conditioning, is principled and shown to be effective on real-world and simulated manipulation tasks.

The reviewers raised a few questions/issues with the initial submission and the authors clearly made a significant effort to address these concerns, through detailed responses to each of the reviewers, additional results, and updates to the paper. Each of the reviewers commented that this effort clarified several of their questions and addressed many of their initial concerns. However, the reviewers remain unconvinced that the amount of domain knowledge assumed for the proposed approach with regards to the environment abstraction and the plan generation is consistent with existing one-shot learning methods. They also find that it is not clear how one would apply the framework to other domains. The AC acknowledges the authors' comparisons to the assumptions made by contemporary one-shot learning methods (e.g., the use of human demonstrations), but finds the arguments not entirely convincing in their current form. This is a promising line of work with good potential, and a revised version of the paper that resolving these remaining issues would provide a solid contribution.

**Summary Of Ac-Reviewer Meeting:**

N/A